# Spatial assessment of probable recharge areas - Investigating the hydrogeological controls of an active deep-seated gravitational slope deformation

Jan Pfeiffer[1,2], Thomas Zieher[1], Jan Schmieder[1,2], Thom Bogaard[3], Martin Rutzinger[2], and Christoph Spötl[4]

[1]Institute for Interdisciplinary Mountain Research, Austrian Academy of Sciences, Innrain 25, 6020 Innsbruck, Austria
[2]Department of Geography, University of Innsbruck, Innrain 52f, 6020 Innsbruck, Austria
[3]Faculty of Civil Engineering and Geosciences, Delft University of Technology, Stevinweg 1, 2628 CN Delft, Netherlands
[4]Institute of Geology, University of Innsbruck, Innrain 52f, 6020 Innsbruck, Austria

**Correspondence:** Jan Pfeiffer (jan.pfeiffer@oeaw.ac.at)

**Abstract.** Continuous and slow-moving deep-seated landslides entail challenges for the effective planning of mitigation strategies aiming at the reduction of landslide movements. Given that the activity of most of these landslides is governed by pore pressure variations within the shear zone, profound knowledge about their hydrogeological control is required. In this context, the present study presents a new approach for the spatial assessment of probable recharge areas to better understand a slope's hydrogeological system. The highly automated geo-statistical approach allows deriving recharge probability maps of groundwater based on stable isotope monitoring and a digital elevation model (DEM). By monitoring stable isotopes in both, groundwater and precipitation, mean elevations of recharge areas can be determined and further constrained in space with the help of the DEM. The approach was applied to the Vögelsberg landslide, an active slab of a deep-seated gravitational slope deformation (DSGSD) in the Watten valley (Tyrol, Austria). Resulting recharge probability maps indicate that shallow groundwater emerging at springs on the landslide recharges between 1100-1500 m a.s.l.. In contrast, groundwater encountered in wells in up to 49 m below the landslide's surface indicates a mean recharge elevation of up to 2200 m a.s.l. matching the highest parts of the catchment. Further inferred proxies, including flow path length, estimated recharge area sizes, and mean transit times of groundwater resulted in a profound understanding of the hydrogeological driver of the landslide. It is shown that the new approach can provide valuable insights into the spatial pattern of probable recharge areas where mitigation measures aiming at reducing groundwater recharge could be most effective.

**Keywords:** Stable water isotopes, landslide, DSGSD, hydrogeology, mean recharge elevation

## 1 Introduction

The variability of slow but continuous movements of large-scale, deep-seated gravitational slope deformations (DSGSDs) pose a serious risk to anthropogenic infrastructure and therefore threaten human needs in mountain areas (Crosta et al., 2013; Cignetti et al., 2020). In many cases, secondary subunits show enhanced activity forced by pore-pressure changes related to

groundwater variations (Lacroix et al., 2020). Mitigation measures aiming at the reduction of the pore pressure provide a promising tool to decrease the activity of moving slopes (Eberhardt et al., 2007; Hofmann and Sausgruber, 2017). In this context, a detailed understanding of the hydrogeological processes controlling pore-pressure variations is an essential requirement

to enhance the effectiveness of mitigation strategies. It is important to know where the landslide-controlling groundwater originates and which subsurface flow path(s) it takes until the water reach the landslide's governing aquifer or the pore-pressure wave within the saturated zone reaches the landslide (Bogaard and Greco, 2016). Only then it is possible to develop solutions to effectively drain a slope in order to reduce a landslide's activity. The performance of nature-based solutions may strongly be enhanced by a complete understanding of a landslide's driver (Kumar et al., 2021b, a)).

Groundwater recharge is controlled by precipitation and/or surface water percolating through the unsaturated zone towards the aquifer. Flow within the aquifer is further controlled by gravity and its hydraulic properties (Welch and Allen, 2014). Various methods can be used to characterise groundwater flow dynamics, storage and recharge areas including (i) hydrochemistry (Cervi et al., 2012), (ii) hydromechanical modelling (Cappa et al., 2004), (iii) reactive and conservative tracer tests (Bogaard et al., 2007; Vallet et al., 2015; Hilberg, 2016; Ronchetti et al., 2020) and, (iv) non-invasive geophysical methods (Jomard et al.,

2007; Siemon et al., 2009; Chalikakis et al., 2011; Zieher et al., 2017; Lajaunie et al., 2019). The use of tracers to track water through the slope's subsurface entails an auspicious way to assess and characterise the hydrogeological forcing of a landslide. Natural and artificial tracers have been widely used to obtain insights into the hydrogeological setting of deep-seated slides (Montety et al., 2007; Vallet et al., 2015; Strauhal et al., 2016; Koltai et al., 2018). Whereas the use of artificial tracers relies on a well-designed experimental setup and assumptions of potential flow mechanisms, the advantage of natural tracers is that

their availability is ubiquitous and tied to natural processes of groundwater recharge (e.g. rainfall and snowmelt) (Binet et al., 2007; Ronchetti et al., 2020).

The stable isotopic composition of water is a widely used conservative tracer applied to a wide range of hydrological studies at different scales and with different objectives. On a global scale, Jasechko et al. (2014) investigated stable isotope ratios in precipitation and groundwater to determine the seasonal differences in groundwater recharge. On a regional scale, Blasch and

Bryson (2007) identified seasonality and the dominating areas of recharge within several subbasins by using stable isotope values of precipitation and groundwater. On a catchment scale, Schmieder et al. (2016), exploited stable isotopic signatures combined with a physically-based snow model for hydrograph separation and quantification of snowmelt contributions to streamflow. Subsequently, for better understanding and quantification of landslide hydrology, stable isotope-supported characterisation of recharge, constraining groundwater flow systems, and estimating mean recharge elevations are proven method-

ologies (Scanlon et al., 2002; Bouchaou et al., 2008; Guglielmi et al., 2002; Mikoš et al., 2004). The underlying principle is the altitude-dependent fractionation of stable hydrogen and oxygen isotopes in precipitation creating a systematic decrease in the abundance of $^{18}$O with increasing elevation (Dansgaard, 1964). Comparing the stable isotopic composition of groundwater sampled at springs or in wells with that in precipitation allows the estimation of mean recharge elevations (Moser and Rauert, 1980; Blasch and Bryson, 2007). Many studies utilised this 'altitude effect' to infer flow paths between identified recharge

elevation and discharge location (e.g. a spring) along a simple 2D transect (Guglielmi et al., 2002; Madritsch and Millen, 2007; Hilberg and Riepler, 2016). Commonly, the result of such studies is a conceptualised hydrogeological model describing the

groundwater flow along the slope.

Guglielmi et al. (2002), for example, investigated the groundwater recharge mechanisms of two different deep-seated landslides in the French Alps. For both landslides, they differentiated a shallow and a deep-seated groundwater flow. A hydrogeological study presented by Madritsch and Millen (2007) evidenced groundwater flow paths along a continuous basal shear zone ranging from the highest elevations within a DSGSD towards its toe. Furthermore, the authors affirmed the importance of having a local $\delta^{18}O$ precipitation altitude gradient for the assessment of recharge areas. This is supported by results presented by Liebminger et al. (2006) who determined distinct differences in the isotopic composition of precipitation on the margin of the Alps compared to inneralpine settings. Cervi et al. (2012) identified the origin of groundwater and assessed deep groundwater inflow into a landslide using a wide range of methods including hydrochemistry and in situ monitoring. Cappa et al. (2004) coupled hydromechanical modelling with long-term hydrochemical monitoring at a large moving rock slope to investigate the influence of location and amount of water infiltration on the landslide's hydromechanical behaviour. The authors of these studies described the hydrogeological mechanisms forcing deep-seated landslides. However, studies employing the monitoring of stable isotopes typically lack locally validated $\delta^{18}O$ altitude gradients based on periodical sampling of precipitation across the respective elevation range (Madritsch and Millen, 2007; Liebminger et al., 2006). Another knowledge gap identified in the literature concerns the area-wide and quantitative analysis of probable recharge areas (Vallet et al., 2015; Ronchetti et al., 2020). The present study addresses these gaps by combining local isotopic precipitation, groundwater time series and coherent geodata to assess areas of groundwater recharge throughout the landslide's catchment area. Resulting maps of probable recharge areas are seen as an opportunity to enhance the understanding of the hydrogeologic drivers of deep-seated landslides required for planning effective drainage measures.

In this study we introduce and apply a novel geo-statistical approach for the 3D delineation of probable groundwater recharge areas. The approach is applied to the deep-seated Vögelsberg landslide in the Watten valley (Tyrol, Austria). This paper is going to (a) present the geo-statistical approach and the required hydrogeological monitoring of groundwater and precipitation, (b) derive the local altitude gradient of $\delta^{18}O$ in precipitation and compare it against regional datasets, (c) identify probable recharge areas of the landslide-controlling groundwater, and (d) establish a conceptual model of the hydrogeological controls of the landslide.

## 2 The Vögelsberg landslide

The north-east facing slope at the entrance of the Watten valley in Tyrol (European Alps, Austria) is affected by a deep-seated gravitational slope deformation (DSGSD) with an approximate N-S extent of 3 km and E-W extent of 1.6 km (Figure 1a). This DSGSD ranges from 750 to 2150 m a.s.l. wherein an active part (ca. 500 x 500 m) is situated on the foot-slope between 750 to 1050 m a.s.l. A N-S striking ridge represents the western and upper boundary of the topographic DSGSD-catchment. The ridge rises from 1100 m in the north to 2200 m in the south. The Wattenbach draining the Watten valley limits the DSGSD towards the east at an elevation between 750 m and 950 m a.s.l. in the study area. The latter belongs to the Austroalpine

Innsbruck Quartzphyllite complex consisting of early Paleozoic greenschist-grade meta-sediments dominated by quartzphyllite

90 (Rockenschaub et al., 2003).

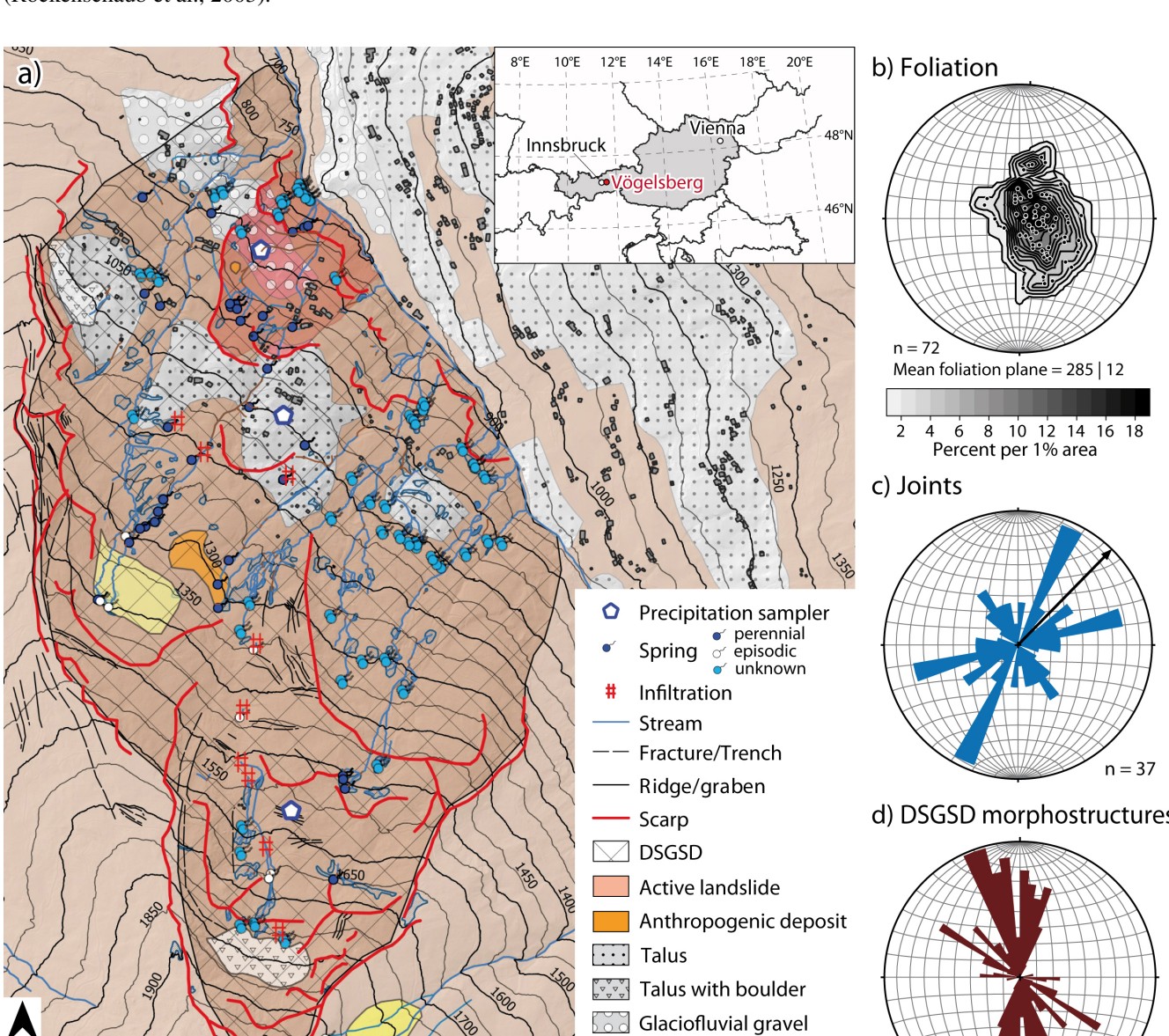

**Figure 1.** Geological map of the study area (a), stereographic contour plot of foliation with pole points (b) and rose plots indicating the orientation of joints (c) and DSGSD morphostructures including scarps, fractures, trenches, ridges and grabens (d) (DTM and spring inventory: Federal state of Tyrol, Geological basemap adapted from GBA).

## 2.1 Hydrogeological characterization

Outcrops within the study area are restricted to quartzphyllites where intercalations of marble layers varying in thickness from a few centimetres to several metres are common. The orientation of the foliation is uniform across the slope and slightly dipping towards WNW (Figure 1b). Joints are abundant, showing a preferred orientation of open and sub-vertical joints along NNE-SSW and ENE-WSW (Figure 1c). Extensional morphological features such as double crested ridges, grabens, trenches, open fractures and scarps of different generations are present along and in proximity to the ridge, marking the upper western boundary of the DSGSD. The orientation of most of these morphostructures is in agreement with the NNW-SSE striking ridge (Figure 1d). Heavily fractured bedrock dominates the ridge. Till and fluvial gravel are present in local morphological depressions on the DSGSD.

The fractured quartzphyllite constitutes the main aquifer providing flow paths. Due to the high density of well-connected fractures this aquifer is assumed to provide isotropic flow similar to porous aquifers (Guglielmi et al., 2002). Thin and patchy Quaternary sediments might act as small local porous aquifers or aquitards depending on their grain size. Shear zones and landslide slip surfaces are assumed to induce a strong directional hydraulic heterogeneity along which the hydrogeological properties deviate strongly from those of the surrounding. While shear zones in quartzphyllite create low-permeability zones, brittle deformation commonly increases permeability along densely fractured shear zones (Binet et al., 2007; Bonzanigo et al., 2007).

The highest springs draining the DSGSD can be found 150 to 350 m below the respective ridge elevation. This is the case at 1200 m a.s.l. for the northernmost part and at 1700 m a.s.l. for the southernmost part of the DSGSD. In some cases, the spring water follows the stream network only for a short distance and infiltrates into the subsurface further downslope. This phenomenon can especially be observed during snowmelt and at the uppermost springs (approx. >1500 m a.s.l.). They tend to fall dry after long periods without precipitation or snowmelt. On the other hand, springs at lower elevations (<1350 m a.s.l.) show continuous discharge. In general, this lower area is characterised by a high density of springs resulting in a dense network of small creeks. The mean perpendicular distance between the 5 creeks draining the DSGSD is 250 m, while the mean spring density is 1 spring per 130 x 130 m.

## 2.2 Hydro-meteorological landslide forcing

In a previous study Pfeiffer et al. (2021) have shown that phases of landslide acceleration occur as a delayed response to long-lasting rainfall and/or snow-melt events. Constant displacement rates of 1-2 cm yr$^{-1}$ prevail most of the year while non-seasonal and extraordinary hydro-meteorological events result in mean rates of up to almost 6 cm yr$^{-1}$. Results of a hydro-climatological model indicate an annual average of 644 mm of water (snowmelt + rainfall - evapotranspiration) in the catchment area of the landslide between 2009 and 2018. Hydro-meteorological events of up to 600 mm of water within 100-200 days triggered phases of accelerated landslide movement. Pfeiffer et al. (2021) also show that the landslide's response time varies depending on the type of water source. While spatially heterogeneous snowmelt induces a relatively fast (0-8 days) response, spatially evenly distributed rainfall entails a delayed (approximately 45 days) response of landslide acceleration. Computed spatially-

distributed response times suggest a spatially varying delay in case of acceleration phases triggered by snowmelt. For example, in early 2019 snowmelt on and right above the landslide occurred only a few days before the respective landslide acceleration. In contrast, in areas above 1700 m snowmelt occurred after the acceleration phase and therefore likely did not contribute to the landslide's short-term dynamics (Pfeiffer et al., 2021).

## 3 Material and Methods

### 3.1 Field work

Detailed hydrogeological mapping by the "Landesgeologie" of the Federal State of Tyrol in 2016 provided valuable information about the location of creeks and springs. Additional field mapping was carried out in 2019 by paying special attention to geological and morphological structures bearing essential information of the subsurface hydrodynamic behaviour. The orientation of foliation and joint surfaces observed at bedrock outcrops was recorded using a geological compass. Morphological structures associated with DSGSD activity (e.g. scarps, counter scarps, extensional fractures and trenches) were mapped in the field and digitized with the help of a shaded relief image derived from a digital terrain model (DTM) based on airborne laser scanning (ALS) by the Federal State of Tyrol.

Precipitation was collected between July 2019 and July 2021 using four Palmex (http://www.rainsampler.com, last accessed 2022-06-07) samplers (Gröning et al., 2012) mounted on a pole and installed at 880 m, 1095 m, 1577 m and 1980 m a.s.l. (Figure 2a and c). Water samples were collected during each field campaign, and the amount of rain and melted snow was recorded. At the end of the winter season (2020-04-15 and 2021-03-30), snow pits were dug at the uppermost location (1980 m a.s.l.) to estimate the snow water equivalent (SWE) by weighting snow samples within a known volume. Snow samples were collected for isotopic analysis.

As an independent reference, monthly precipitation data from the Austrian Network for Isotopes in Precipitation (ANIP) stations Innsbruck (580 m a.s.l. and 18 km from the study area) and Patscherkofel (2245 m a.s.l. and 13 km from the study area) were used, providing a complete time series from 1988-09-01 to 2001-09-01 (Figure 2a). Hydrogeological monitoring campaigns were carried out from October 2018 until June 2020. Based on the hydrogeological inventory provided by the Federal State of Tyrol (Figure 1a), 35 measurements points fulfil the demands to be part of a temporally condensed measurement setup. Selection of measurement points was done based on the following criteria: measurement points are accessible and permitted to be accessed by the owner during the monitoring period, natural water outlets are effectively measurable without disturbance of the surrounding environment, and measurement points intersect the assumed area of potential landslide influence. The assumed area of potential landslide influence covers the lowest and highest part of the DSGSD and is grouped into three elevation bands (Figure 2b). Measurement point designation is based on the discharge elevation and prefixed acronym indicating the elevation band. Measurement points L01 – L10 intersect with the sections of the lower slope and the area of the active landslide (<1000 m a.s.l.). Measurement points M11 – M31 intersect with the middle slope section (1000 – 1500 m a.s.l.) and U32-U35 with the upper slope (>1500 m a.s.l.).

Multitemporal measurements of discharge (Q in $l\,sec^{-1}$), water temperature (T in °C) and electrical conductivity (EC in $\mu S\,cm^{-1}$)

were conducted at 35 measurement points including natural springs (n=14), housed springs (n=17) and drainages (n=4) within the DSGSD (Figure 2b). Housed springs are structurally supported groundwater outlets and relevant for residents' water supply. Natural springs are mostly diffuse zones of groundwater exfiltration. Discharge was measured using a bucket and a stop watch. EC and T were measured in the field using a WTW Cond 3310 device (Figure 2d). Water samples for stable isotope analyses were periodically taken from selected springs and drainages. Furthermore, T and EC measurements as well as multiple water samples were obtained from two groundwater wells (KB1 and KB2) located on the active part of the landslide. The measurements and samples were taken at various depths using a sampling probe on 2018-11-29, 2019-04-16 and 2019-08-13. Screens allowing groundwater to enter the wells are installed at 16 to 49 m depth in KB1 and at 21 to 39 m depth in KB2. Well measurements and sampling was done at constant intervals from the piezometric height towards the bottom of the well. Both wells are equipped with piezometers recording the groundwater level and are operated by the Austrian Service for Torrent and Avalanche Control. Table1 summarises acquired and utilised field data and their measurement properties.

Precipitation samples of snow, rain, and groundwater were analysed regarding their oxygen isotopic composition. Samples were stored at 4°C before being analysed with a Picarro L2140-i cavity ring down spectroscopy analyser following the procedures outlined by van Geldern and Barth (2012). Results are reported in per mill (‰) against the Vienna Standard Mean Ocean Water (VSMOW). Calibration of the instrument was accomplished using VSMOW2, GISP2, and SLAP standards. The long-term analytical precision of measurements is 0.1 ‰ (1 $\sigma$). Acquired hydrogeological monitoring data serves as the basis for applying the subsequently presented approach for assessing probable recharge areas (Figure 3a).

### 3.2 Estimation of inverse transit time proxy

The inverse transit time proxy (ITTP) proposed by Tetzlaff et al. (2009) was used to estimate groundwater transit times. The ITTP is defined as the ratio of the standard deviation of $\delta^{18}O$ in groundwater to the standard deviation of $\delta^{18}O$ in precipitation. The proxy reflects the degree of attenuation of annually varying $\delta^{18}O$ of precipitation in groundwater where a lower ratio indicates a higher attenuation and a longer transit time. The ITTP was calculated for every groundwater sample where at least five multi-temporal $\delta^{18}O$ measurements were available for calculating the standard deviation. Standard deviations from multitemporal $\delta^{18}O$ ratios in precipitation were calculated from ANIP and Vögelsberg samples.

### 3.3 Estimation of probable recharge elevations

Mean recharge elevations were estimated for each spring and groundwater sample using the altitude effect of oxygen isotopes in precipitation (Moser and Rauert, 1980; Clark and Fritz, 1997; Mook, 2006). This effect has been applied in several landslide case studies and requires a precise estimation of the local $\delta^{18}O$ altitude gradient of precipitation which can be described by the linear model's coefficients $\beta$ (slope) and $\alpha$ (intercept) (Madritsch and Millen, 2007; Vallet et al., 2015; Hilberg and Riepler, 2016). Two local annual $\delta^{18}O$ altitude gradients of precipitation were derived for the time spans 2019/07 to 2020/06 and 2020/06 to 2021/07 (referred to as 19/20 and 20/21 Vögelsberg gradients). Furthermore, 13 annual altitude gradients were computed from the ANIP $\delta^{18}O$ time series covering the period from 1988/09 to 2001/09. The ANIP altitude gradients show good agreement with the locally derived 19/20 and 20/21 Vögelsberg gradients (section 4.3). Therefore, all $\delta^{18}O$ time series

**Table 1.** Overview of acquired and used groundwater (GW) and precipitation (P) data.

| Water type | Measurement category | Parameters | Period | Measurement interval | Source | Measured points | Elevation [m] a.s.l. |
|---|---|---|---|---|---|---|---|
| GW | spring housed | Q, EC, T, $\delta^{18}$O | Oct. 2018 - Jun. 2020 | approx. 1.5 months | this study | 17 | 1000-1500 |
| GW | spring natural | Q, EC, T, $\delta^{18}$O | Oct. 2018 - Jun. 2020 | approx. 1.5 months | this study | 14 | 815-1640 |
| GW | drainage | Q, EC, T, $\delta^{18}$O | Oct. 2018 - Jun. 2020 | approx. 1.5 months | this study | 4 | 880-1030 |
| GW | well | EC, T, $\delta^{18}$O | Nov. 2018-Aug. 2019 | approx. 4 months | this study | 2 | 841-899 |
| P | rain & snow | SWE, $\delta^{18}$O | Jul 2019 - Jul 2021 | approx. 1.5 months | this study | 4 | 880, 1095, 1577, 1980 |
| P | snow (snow pit) | SWE, $\delta^{18}$O | Apr. 2020 - Mar 2021 | once a year | this study | 1 | 1980 |
| P | rain & snow | SWE, $\delta^{18}$O | Sep. 1988- Sep. 2001 | 1 month | ANIP | 2 | 580, 2245 |

were used to establish a linear model of the $\delta^{18}$O ratio against elevation, considering weights according to the precipitation amount. Furthermore, the probability distribution and respective 95% confidence bands were computed as lower and upper boundaries of this long-term gradient, representing the inter-annual variability of $\delta^{18}$O in precipitation.

For subsequently estimating the mean recharge elevation (MRE) of a groundwater sample, the respective isotope ratio $\delta^{18}O_{GW}$ was passed to the linear equation of the long-term $\delta^{18}$O altitude gradient of precipitation:

$$MRE = \alpha + \beta * \delta^{18}O_{GW} \tag{1}$$

where $\alpha$ and $\beta$ are the coefficients of the derived long-term gradient. By considering the gradient's level of confidence a probability distribution of mean recharge elevations was deduced for each groundwater sample (Figure 3b). This probability distribution was then scaled to a range of 0 to 1 in with 0 represents a low and 1 a high recharge elevation probability.

### 3.4 Geodata-supported approach for the derivation of probable recharge areas

The recharge elevation provides essential information about a slope's governing hydrogeological system (Guglielmi et al., 2002). Furthermore, the probability distribution of recharge elevation enhances the potential for further analysis towards a spatial assessment of probable recharge areas. In this context we present a new method to constrain probable recharge areas by transferring the stable isotope-based recharge elevations into the third dimension. This requires a digital terrain model

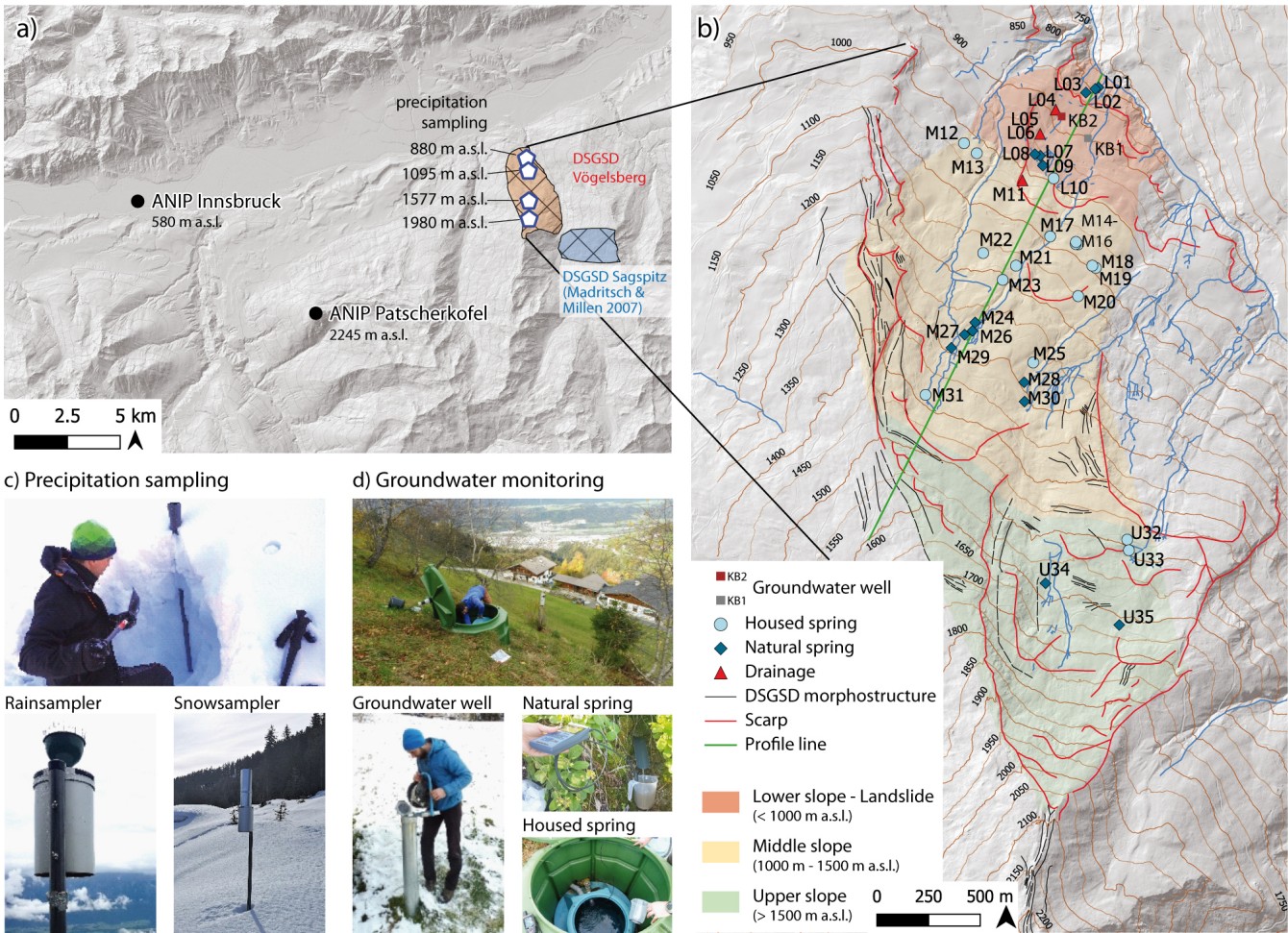

**Figure 2.** Overview of the hydrogeological monitoring. a) Study area and nearby Austrian Network for Isotopes in Precipitation (ANIP) stations Innsbruck and Patscherkofel; b) map of monitoring sites, locations of measurement points selected for monitoring; c) and d) fieldwork impressions of precipitation and groundwater sampling (DTM source: Federal State of Tyrol).

(DTM) representing the ground surface elevation of each raster cell within a catchment, the xyz-coordinates of groundwater sampling points and the respective probability distributions of recharge elevations. These elevation distributions (section 3.3) were first compared against the elevation information of the DTM. An elevation-ranked recharge probability map was compiled by transferring the probability values to the respective recharge elevations (Figure 3c). Raster cells with the value of 1 represent the highest probability and raster cells with the value of 0 the lowest probability.

In order to discriminate between potential recharge areas at the same elevation but differing in proximity to the discharge location, a second distance-dependent indicator was introduced. Based on the assumption that water flow along the shorter and direct path between recharge and discharge location is more likely, 3D distances were stepwise calculated between every

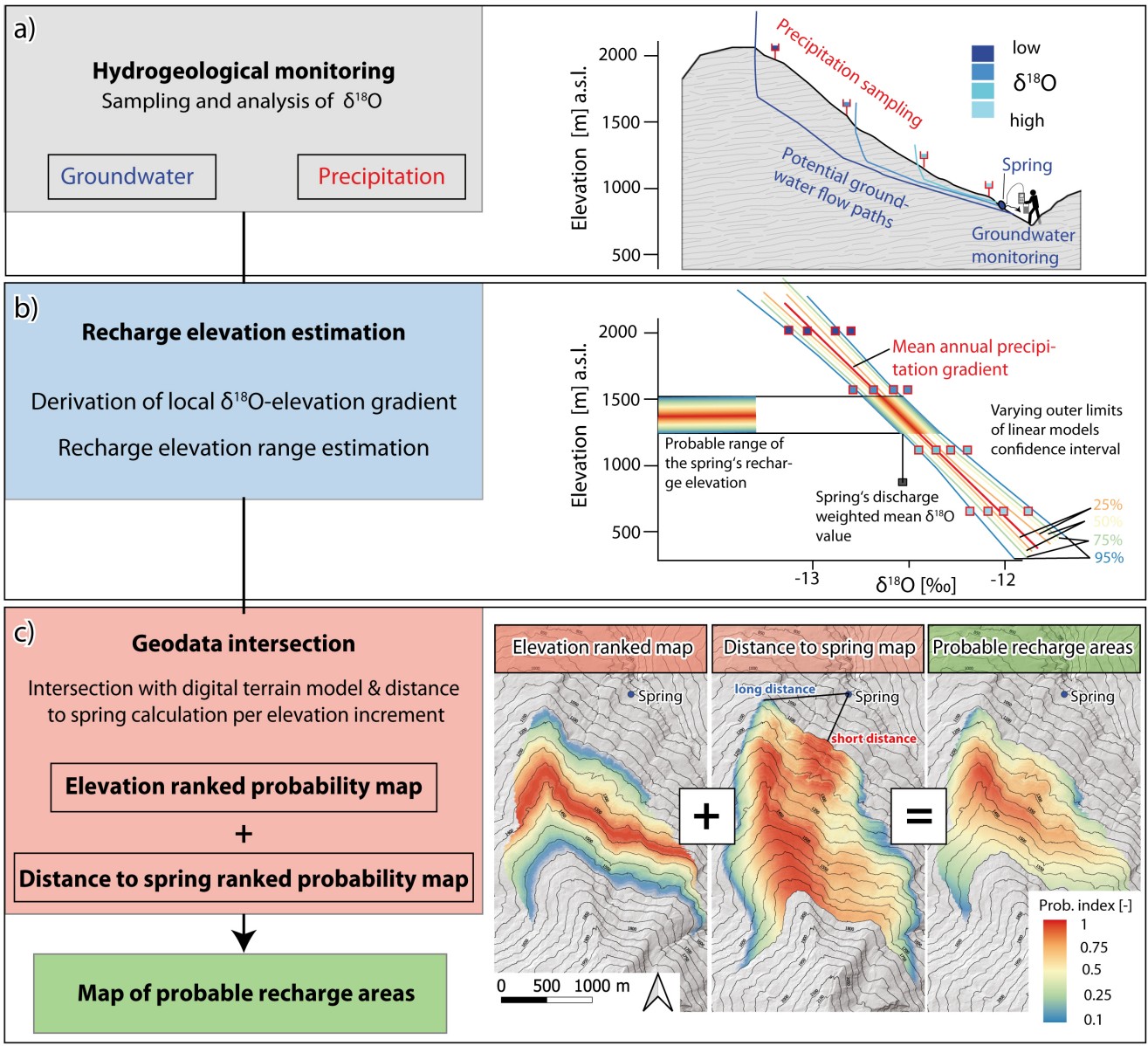

**Figure 3.** Proposed workflow for deriving maps of probable recharge areas including a) Hydrogeological monitoring b) recharge elevation estimation and c) geodata intersection (DTM source: Federal State of Tyrol).

raster cell and the discharge location. 3D distances of the same elevation step were ranked by scaling the distances to a spring to a range of 0 to 1, the distance-to-spring index. A value of 0 represents the maximum distance and a value of 1 represents the minimum distance to the respective spring per elevation band. Introducing a maximum limit defined by a multiplier of

the minimum distance to the spring per elevation class yields a fan-shaped area of flow path distance dependent recharge probability.

The combination of the elevation-ranked recharge probability map and the map showing the distance to spring index results in a map of probable recharge areas. This is accomplished by adding the two maps and applying a 0-1 scaling. The resulting recharge probability maps implicate the unique possibility to further infer potential groundwater flow-path lengths specified as the 3D

distance between the maximum value of the recharge probability map and its discharge location. Aggregation of individual recharge probability maps using mean values creates a single recharge probability map covering recharge information of all investigated groundwater.

We further used discharge measurements (Q) for each spring to estimate the size of the potential recharge area (A) provided that hydroclimatological input data (I) are available. In case only a fraction of the water infiltrates into the subsurface, an additional

coefficient $I_c$ is introduced. $I_c$ varies between 0 and 1, according to 0% and 100% infiltration, respectively (USDA-SCS, 1986):

$$A = \frac{Q}{I * I_c} \tag{2}$$

The estimated size of the recharge area was transferred to the recharge probability map indicating the concordant area with

highest probability values. The described workflow for the assessment of probable recharge areas was automated using the R programming language (R Core Team, 2021). A DTM in the same coordinate reference system as the location of the groundwater discharge locations and in a 5 m spatial resolution was used within the workflow.

## 4   Results

### 4.1   Electrical conductivity and water temperature

The EC of selected springs and drainages shows variations in space and time. A spatial variation between 80 - 600 $\mu$S cm$^{-1}$ of absolute EC-values was observed at different measurement points. Lower values in the order of 80 - 150 $\mu$S cm$^{-1}$ are commonly observed at the uppermost springs, while samples from lower elevations show values between 400 - 600 $\mu$S cm$^{-1}$ (Figure 4a). Measurements of EC along the KB1 well show values up to 395 $\mu$S cm$^{-1}$ for the upper part and 388 $\mu$S cm$^{-1}$ for the lower part (49 m below the surface). The KB2 well shows larger variations with 550 $\mu$S cm$^{-1}$ close to the surface and 375 $\mu$S cm$^{-1}$ at 39 m

depth (Figure 4d). The high values of 550 $\mu$S cm$^{-1}$ are in agreement with data from nearby drainages.

The annual range of EC at individual measurement points is in the range from 5 to 132 $\mu$S cm$^{-1}$. For the majority of measurement points (31 out of 35) the temporal variability is below 66 $\mu$S cm$^{-1}$. In general, minimum values were observed in spring and early summer, whereas maximum values occur in autumn and winter. No distinct temporal variability of EC was observed within the KB1 well (variability < 2 $\mu$S cm$^{-1}$). EC values between the measurement campaigns in KB2 on the other hand vary

up to 25 $\mu$S cm$^{-1}$ at comparable depths.

The mean T of groundwater ranges between 8°C and 10°C at lower sites and 5 to 6°C at higher locations (Figure 4b). A

constant T of 8.7°C was measured within the lowest 39 m of the KB1 well. A tendency towards higher Ts of up to 9.3°C is observable within the uppermost part of this well. Temperatures in the KB2 well show similar values between 8.7 and 9°C. The T of the wells is in general agreement with that of nearby springs.

The water T of most of the springs follows a seasonal pattern with higher values during summer and lower values during winter. The difference between annual minima and maxima reaches up to 5°C. The water T within the wells is constant over time with variation below 0.2°C in KB2 and no measurable differences in KB1.

## 4.2 Discharge, inferred recharge area sizes, piezometric height and landslide movement

The annual mean discharge of all measurement points is on average $0.2\,l\,sec^{-1}$. The majority of them (34 out of 35) show mean
values of below $1.0\,l\,sec^{-1}$. Respective annual means indicate recharge area sizes ranging from about 300 m² to 250,000 m² for the individual discharge locations and variable $I_c$,(Eq 2) (Figure 5).

Generally low discharge was observed in autumn 2018 and in summer 2019. Comparably high discharge was measured in spring and early summer 2019 and coincided with a period of intense snowmelt which led to landslide accelerations (Pfeiffer et al., 2021). Within this period, maximum discharge at springs located at lower elevations (e.g. spring M14 at 1090 m a.s.l.)
occur up to almost three months earlier than the maximum discharge of springs located at higher elevations (e.g. spring M31 at 1395 m a.sl.). This phenomenon is evidenced by comparing time-series of scaled spring discharge values (Figure 6). The temporal variability of discharge at springs at lower elevations and closer to the active discharge are in conformity with time-series of piezometric heights. Piezometric heights measured in KB1 show a temporal variability of 1 m between 2018 and 2020 and conformity with a time-series of mean displacement rates (Figure 6). Phases of accelerated landslide movements therefore
coincide with a higher piezometric level and higher discharge at springs close to the landslide. The accelerated landslide movements during the period of higher water levels and increased spring discharge in early 2019 were the response to intensive snowmelt as shown by a physically-based snow model (Pfeiffer et al., 2021). The aquifer response to this, subsequent snowmelt and summer precipitation events is indicated by the comparison of respective groundwater time series with precipitation time series at the close-by Patscherkofel station (Figure 6).

## 270 4.3 $\delta^{18}O$ in precipitation

Maximum inter-annual variations of the stable isotopic composition of precipitation were observed with $\delta^{18}O$ ratios between -3 to -20‰ at the uppermost precipitation sampling location (Figure 7b). Furthermore, the precipitation samples show a distinct seasonal pattern throughout the year with maximum values of $\delta^{18}O$ occurring in summer and autumn, and minimum values in winter and spring.
Analysis of the stable isotopic composition of precipitation sampled at different elevations within the study area indicate a clear decrease of $\delta^{18}O$ with increasing elevation. Whereas the annual $\delta^{18}O$ ratios weighted by the respective amount of water sampled at the uppermost (1980 m) station are between -13.19‰ and -13.71‰, values at the lowermost station (880 m a.s.l.), are between -11.29‰ and –11.31‰ (Figure 8a). Time series of annually weighted $\delta^{18}O$ in precipitation recorded at ANIP's Patscherkofel station (2245 m) range between -12.28‰ to -14.18‰ and at the Innsbruck station (580m) between -10.11‰

**Table 2.** Statistics of physico-chemical spring parameters.

| Sample | Elevation [m] a.s.l. | Discharge [l*sec$^{-1}$] Mean | $\sigma$ | n | Electrical conductivity [$\mu$S*cm$^{-1}$] Mean | $\sigma$ | n | Temperature [°C] Mean | $\sigma$ | n | $\delta^{18}$O [‰] Mean | $\sigma$ | n |
|---|---|---|---|---|---|---|---|---|---|---|---|---|---|
| L01 | 815 | 0.06 | 0.02 | 11 | 461.45 | 13.60 | 11 | 8.30 | 1.08 | 11 | -11.62 | 0.07 | 3 |
| L02 | 824 | 0.81 | 0.39 | 11 | 469.77 | 13.26 | 13 | 8.38 | 0.69 | 13 | -11.67 | 0.16 | 11 |
| L03 | 844 | 0.06 | 0.03 | 12 | 476.36 | 6.43 | 14 | 9.23 | 1.23 | 14 | -11.90 | 0.10 | 11 |
| L04 | 881 | 0.09 | 0.02 | 6 | 558.17 | 31.85 | 6 | 9.17 | 2.46 | 6 | -12.04 | - | 1 |
| L05 | 932 | 0.04 | 0.02 | 11 | 374.18 | 13.24 | 11 | 8.55 | 1.79 | 11 | -11.63 | 0.07 | 6 |
| L06 | 932 | 0.65 | 0.23 | 11 | 433.91 | 14.06 | 11 | 9.17 | 2.30 | 11 | -11.67 | 0.22 | 7 |
| L07 | 975 | 0.01 | 0.01 | 9 | 381.18 | 29.86 | 11 | 8.05 | 1.64 | 11 | -11.80 | 0.08 | 5 |
| L08 | 979 | 0.01 | 0.00 | 8 | 365.75 | 18.01 | 8 | 7.33 | 1.67 | 8 | -11.67 | 0.04 | 4 |
| L09 | 992 | 0.02 | 0.02 | 16 | 369.44 | 14.65 | 17 | 8.13 | 1.97 | 17 | -11.81 | 0.07 | 11 |
| L10 | 999 | - | - | - | 406.20 | 25.46 | 5 | 6.96 | 1.06 | 5 | -11.96 | 0.09 | 4 |
| M11 | 1031 | 0.13 | 0.04 | 8 | 289.75 | 5.28 | 8 | 8.35 | 1.80 | 8 | -11.57 | 0.12 | 3 |
| M12 | 1032 | 0.64 | 0.07 | 10 | 227.90 | 6.35 | 10 | 8.18 | 0.55 | 10 | -11.67 | 0.06 | 5 |
| M13 | 1034 | 0.30 | 0.12 | 9 | 224.78 | 6.57 | 9 | 8.02 | 0.73 | 9 | -11.70 | 0.15 | 6 |
| M16 | 1089 | 0.06 | 0.06 | 6 | 273.00 | 21.67 | 6 | 8.30 | 2.05 | 6 | -11.80 | 0.15 | 2 |
| M14 | 1089 | 0.21 | 0.04 | 16 | 270.07 | 13.09 | 15 | 8.06 | 0.77 | 15 | -11.79 | 0.06 | 11 |
| M15 | 1089 | 0.08 | 0.09 | 13 | 265.31 | 18.87 | 13 | 8.84 | 2.21 | 13 | -11.82 | 0.16 | 2 |
| M17 | 1092 | 0.11 | 0.03 | 7 | 249.57 | 5.68 | 7 | 8.33 | 2.27 | 7 | -11.63 | 0.18 | 6 |
| M18 | 1099 | 0.26 | 0.14 | 15 | 328.73 | 14.49 | 15 | 7.95 | 0.93 | 15 | -11.88 | 0.10 | 10 |
| M19 | 1099 | 0.05 | 0.03 | 7 | 334.43 | 12.61 | 7 | 6.86 | 1.48 | 7 | -11.91 | 0.01 | 2 |
| M20 | 1160 | 0.37 | 0.14 | 9 | 305.22 | 9.43 | 9 | 7.17 | 0.81 | 9 | -11.97 | 0.08 | 8 |
| M21 | 1173 | 0.08 | 0.00 | 6 | 185.31 | 14.15 | 7 | 7.07 | 2.15 | 7 | -11.82 | 0.12 | 4 |
| M22 | 1181 | 0.01 | 0.00 | 12 | 182.75 | 3.73 | 14 | 6.45 | 1.90 | 14 | -11.82 | 0.10 | 7 |
| M23 | 1222 | 0.42 | 0.24 | 13 | 184.98 | 12.37 | 13 | 6.03 | 0.60 | 12 | -11.79 | 0.25 | 12 |
| M24 | 1260 | 0.09 | 0.03 | 15 | 196.60 | 7.42 | 15 | 5.99 | 0.76 | 15 | -11.68 | 0.14 | 11 |
| M25 | 1268 | 0.09 | 0.01 | 12 | 387.67 | 37.60 | 12 | 5.78 | 1.24 | 12 | -11.77 | 0.08 | 8 |
| M26 | 1272 | 0.06 | 0.03 | 14 | 89.64 | 2.73 | 14 | 5.99 | 1.02 | 14 | -11.66 | 0.18 | 11 |
| M27 | 1277 | 0.14 | 0.06 | 14 | 121.91 | 4.08 | 14 | 6.21 | 1.09 | 14 | -11.68 | 0.17 | 6 |
| M28 | 1297 | 0.07 | 0.03 | 13 | 139.46 | 15.54 | 14 | 5.97 | 0.75 | 14 | -11.59 | 0.20 | 8 |
| M29 | 1305 | 0.08 | 0.10 | 16 | 157.38 | 17.52 | 18 | 5.86 | 0.96 | 17 | -11.82 | 0.31 | 12 |
| M30 | 1307 | 0.05 | 0.02 | 14 | 292.47 | 9.96 | 15 | 6.05 | 1.36 | 15 | -11.86 | 0.10 | 7 |
| M31 | 1395 | 0.26 | 0.25 | 15 | 124.57 | 21.11 | 15 | 5.33 | 1.01 | 14 | -11.55 | 0.19 | 12 |
| U32 | 1496 | 0.45 | 0.18 | 7 | 168.17 | 1.91 | 6 | 7.35 | 2.69 | 6 | -12.58 | 0.27 | 2 |
| U33 | 1501 | 1.29 | 0.88 | 12 | 181.67 | 5.17 | 10 | 5.24 | 1.18 | 9 | -12.46 | 0.10 | 8 |
| U34 | 1585 | 0.89 | 1.32 | 10 | 135.74 | 8.72 | 12 | 5.81 | 3.43 | 11 | -12.77 | 0.21 | 8 |
| U35 | 1640 | 0.10 | 0.07 | 6 | 93.88 | 22.17 | 6 | 9.40 | 1.90 | 5 | -12.53 | 0.24 | 2 |

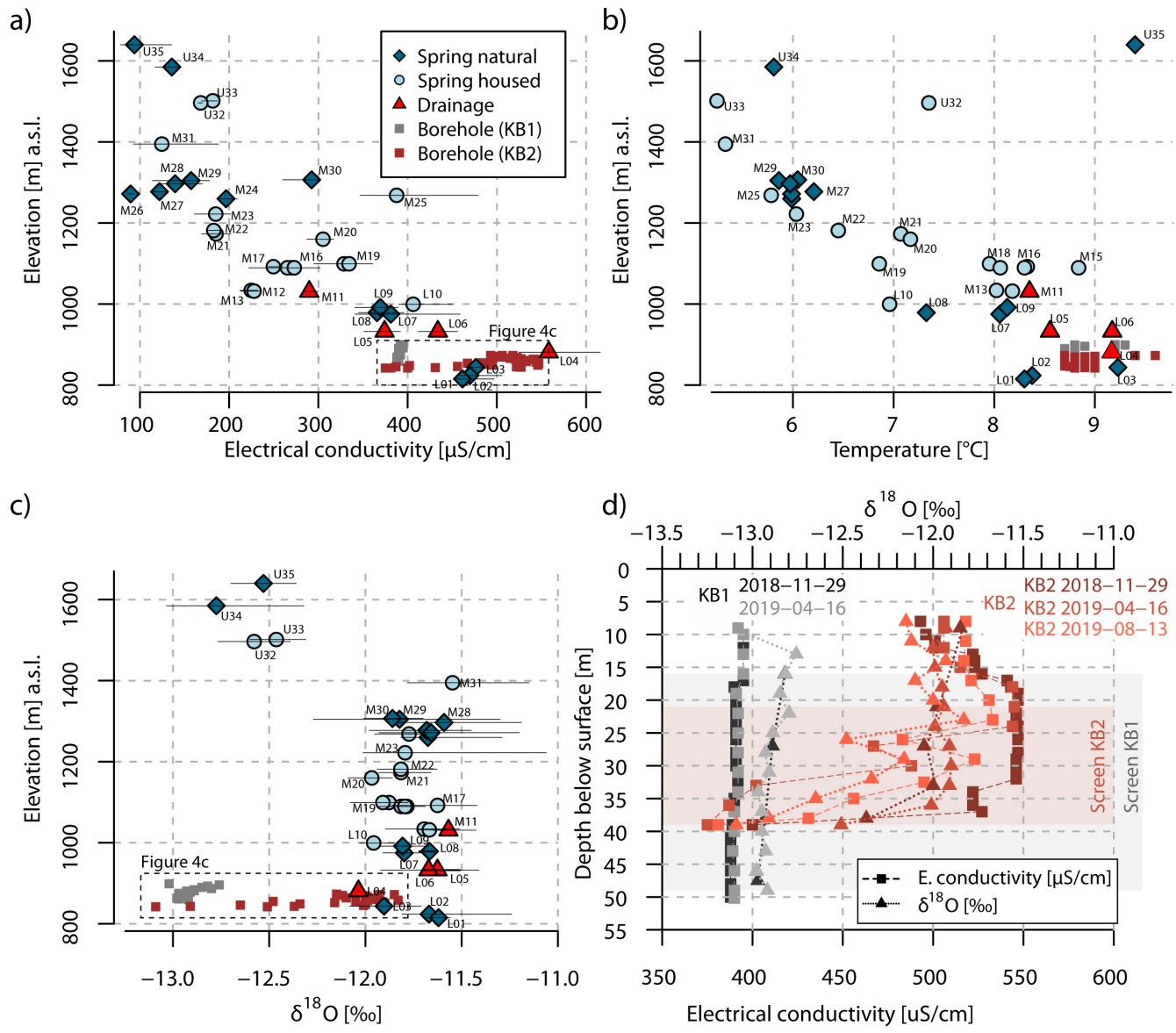

**Figure 4.** Diagrams showing mean values of EC (a), T (b) and $\delta^{18}$O (c) of monitored springs and boreholes. Thin black lines in (a) and (c) indicate the observed range between minimum and maximum values. EC and $\delta^{18}$O measured at different depths in the KB1 and KB2 wells are shown in (d). The groundwater level during sampling can be obtained from Figure 7.

and -12.16‰ within the 13-year record. Linear regressions of all on-site sampled data points result in altitude gradients of 0.21‰ per 100m (20/21) and 0.16‰ per 100 m (19/20). Values of on-site $\delta^{18}$O were extrapolated, utilising derived gradients. Resulting values indicate conformity of the on-site data with respective ANIP reference data (Figure 8b).

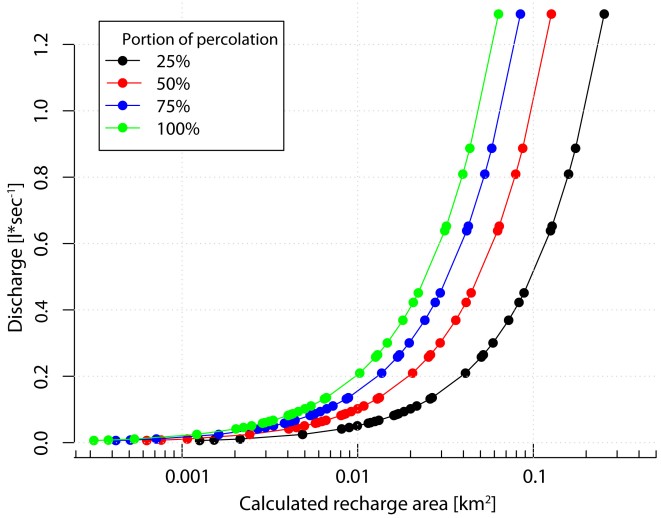

**Figure 5.** Estimated size of recharge area based on average discharge and annual water input of 643 mm (Pfeiffer et al., 2021). The recharge area size was derived from equation 2 for four different infiltration ratios $I_c$ (25%, 50%, 75% and 100%).

Thirteen annual gradients were derived from the ANIP time series showing a range between 0.21‰ per 100m and 0.08‰ per 100m (Figure 8c). The 95% confidence interval (Figure 8a) of ANIP $\delta^{18}$O ratios indicates a range of -13.30‰ to -12.57‰ for the uppermost on-site sampling station at 1980 m a.s.l. and -11.25‰ to -10.54‰ for the lowermost on-site station at 880 m a.s.l.. The calculated mean annual $\delta^{18}$O values for the Vögelsberg sampling stations 880 m, 1095 m and 1577 m a.s.l. are within the determined range of the 95%-confidence from the ANIP-reference gradients. Only the $\delta^{18}$O values for the uppermost sampling location (1980 m a.s.l.) from 20/21 lie below the 95% confidence limit. Consequently, the slope of the fitted linear model of -0.21‰ per 100m representing the altitude gradient for the same period is exceptionally high compared to the long-term average of -0.14‰ per 100m (Figure 8c). The slope of the Vögelsberg gradient for the sampling period 19/20 (-0.16‰ per 100 m), however, is very close to the long-term mean (-0.14‰ per 100m).

## 4.4  $\delta^{18}$O in groundwater

The isotopic composition of groundwater samples was analysed for the period from 2018/11 to 2020/06. Within this period the mean $\delta^{18}$O ratios of drainages and springs below 1450 m a.s.l. range from -12.0 to -11.5‰. No distinct differences in the isotopic composition of the water within the wells were observed between the two (KB1) respective three (KB2) measurement campaigns. Springs located above 1450 m a.s.l. show lower mean values of about -12.5‰. The lowest values (-13‰) were measured in the KB1 well. Samples from this well show more constant $\delta^{18}$O ratios along the vertical profile compared to KB2. KB2 on the other hand shows a similar isotopic composition as the surrounding springs (approx. -12‰) within the upper part of the well and converges towards lower $\delta^{18}$O ratios (-13‰) deeper down (Figure 4d).

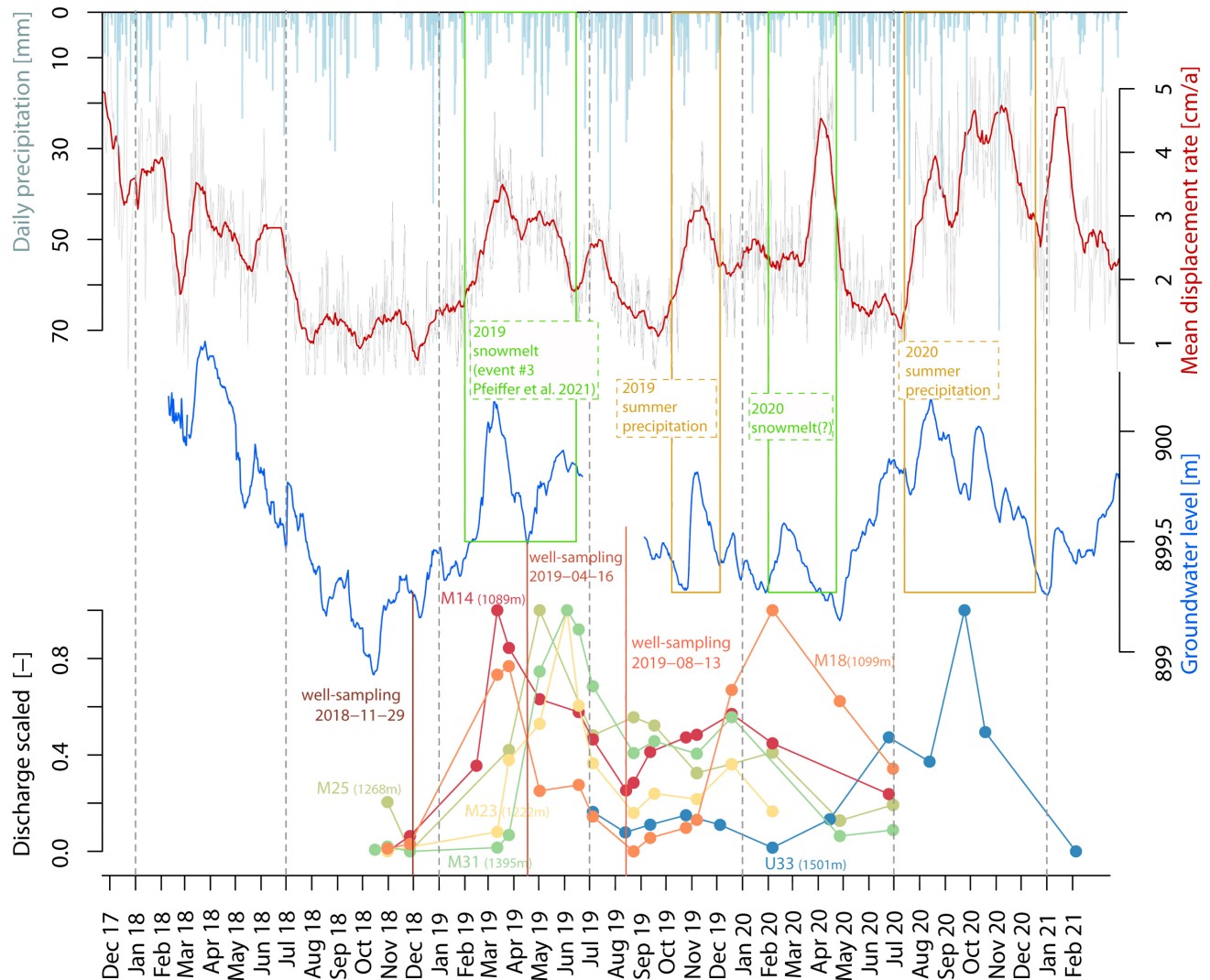

**Figure 6.** Time series of daily precipitation at Patscherkofel (source: ZAMG), mean displacement rate (source: Federal State of Tyrol), groundwater level in well KB1 (source: Austrian Service for Torrent and Avalanche Control) and discharge of selected housed springs. Discharge was scaled between 0 and 1 for better comparability of the different springs.

The temporal variability expressed as the difference between $\delta^{18}$O maxima and minima is less than 1‰ for all water samples (Figure 7a). Springs discharging at high- and mid-elevations (1200-1700 m a.s.l.) show a higher temporal variability (average: 0.6‰) than springs located at lower parts of the slope and the active landslide area (average: 0.2‰).

Considering both, $\delta^{18}$O and $\delta^{2}$H values, the groundwater samples are aligned along the precipitation data and therefore agree with the local meteoric water line. All groundwater samples are evenly surrounded by higher and lower $\delta^{18}$O and $\delta^{2}$H precipitation values indicating a balanced seasonal recharge (Figure 7c).


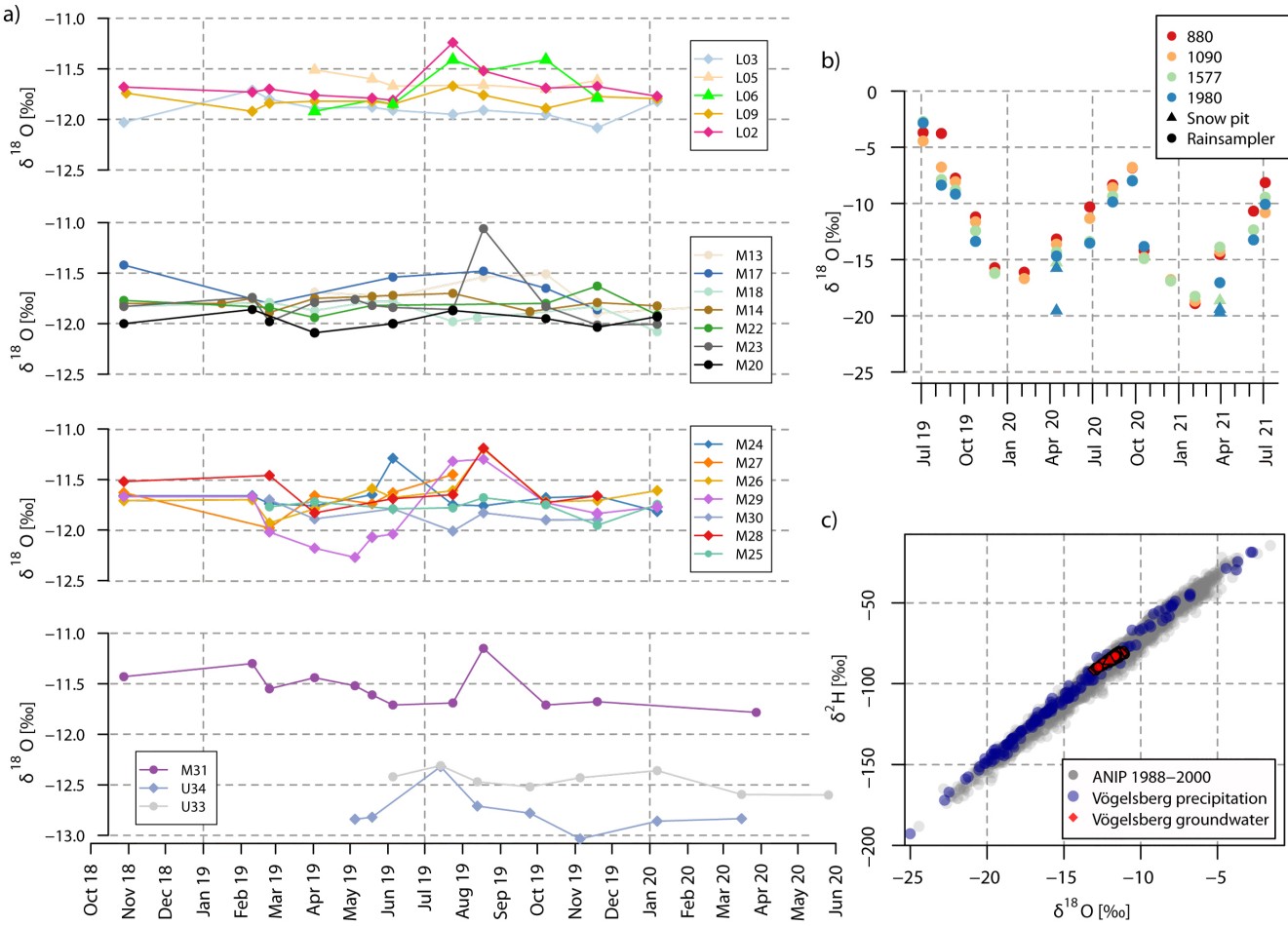

**Figure 7.** Time series of $\delta^{18}O$ of selected springs (a), time series of $\delta^{18}O$ in precipitation (b) and $\delta^{18}O$ vs. $\delta^2H$ in precipitation (ANIP and Vögelsberg) and groundwater (Vögelsberg) (c).

### 4.5 Inverse transit time proxy

The seasonal amplitude of $\delta^{18}O$ in precipitation in Alpine regions shows differences of up to 20‰ between monthly maxima and minima (Liebminger et al., 2006). Time-series of $\delta^{18}O$ in groundwater within the study area on the other hand indicate a
strong attenuation of this input signal in the subsurface. The level of attenuation is reflected by the ITTP after Tetzlaff et al. (2009). Calculated ITTPs for all groundwater samples with more than 5 measurements show standard deviation ratios below 0.1. According to Tetzlaff et al. (2009), who compared ITTP values and published mean transit times (e.g., McGuire et al., 2005; Rodgers et al., 2005a, b; Tetzlaff et al., 2007), a ratio below 0.1 indicates transit times of at least half a year. Groundwater samples in the study area reach ITTP values indicating transit times of up to 3.5 years.

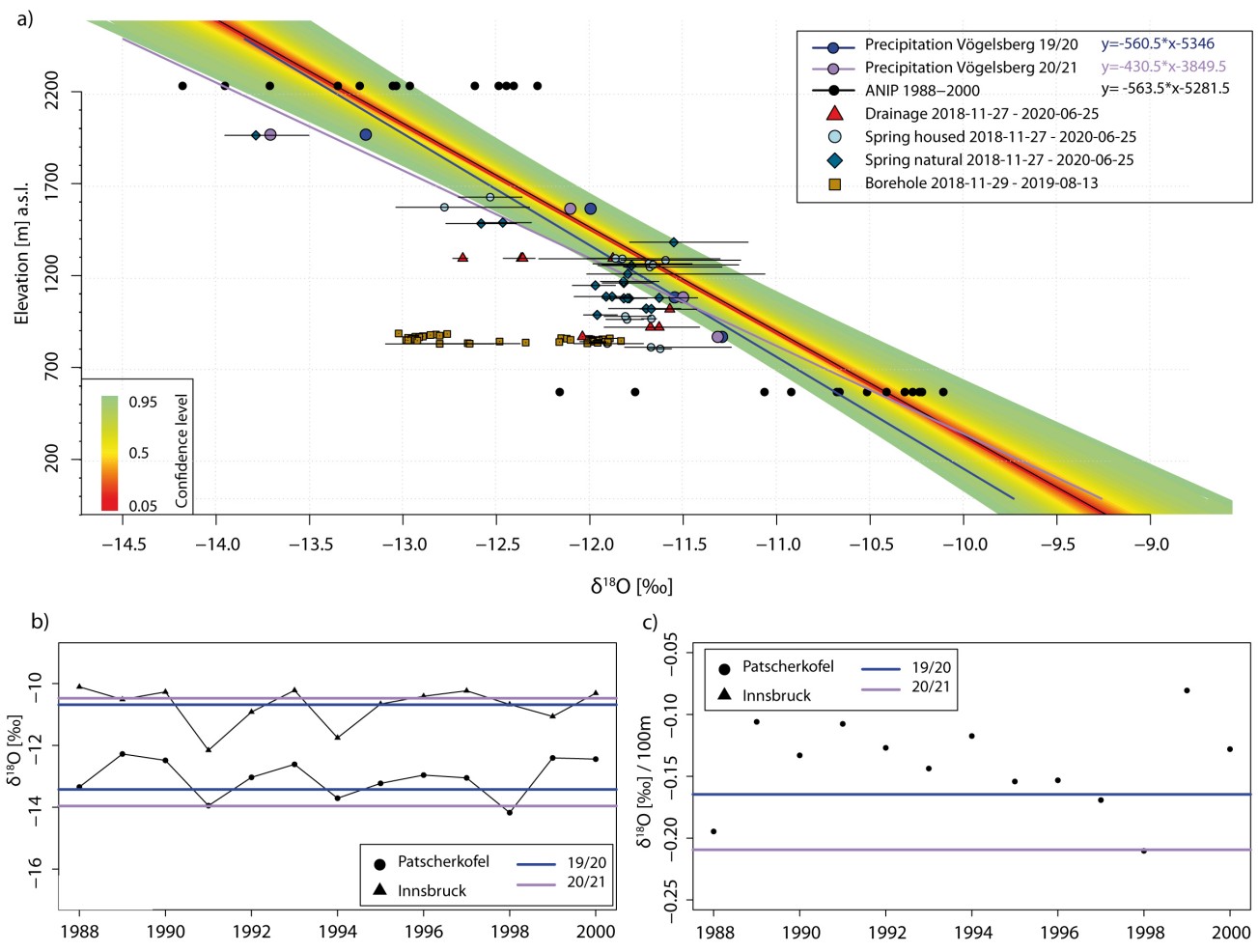

**Figure 8.** $\delta^{18}$O values and elevation of precipitation and groundwater samples. a) Precipitation gradients, confidence interval and groundwater samples. b) Time-series of amount-weighted annual $\delta^{18}$O in precipitation at ANIP stations Patscherkofel and Innsbruck (data retrieved from: https://wasser.umweltbundesamt.at/h2odb/, last accessed 2021-12-20). Annual $\delta^{18}$O values (extrapolated to the corresponding ANIP station elevation) of the study are shown by horizontal lines. c) Time-series of annual $\delta$18O precipitation gradients.

## 4.6 Recharge probability map

$\delta^{18}$O ratios from springs below 1450 m a.s.l. and in the active landslide (750 to 1050 m a.s.l) indicate mean recharge elevations between 1000 m to 1650 m a.s.l. (Figure 8a). Samples from the wells with values as low as -13‰ on the other hand indicate mean recharge elevations of up to 2200 m a.s.l. representing the highest and southernmost parts of the DSGSD. Differentiation of individual recharge areas was accomplished using the geodata-supported approach (section 3.4). Spatial patterns of recharge areas become apparent by comparing recharge probability maps obtained for each discharge location. Differences can be

observed by comparing recharge maps of two springs located at approximately 840 m and 820 m a.s.l. at a distance of 45 m. While one spring (L03) has its recharge area in the highly fractured more northern part of the ridge at an average elevation of 1350 m a.s.l., the other springs (L01 and L02) receive water recharging closer to the landslide at 1170 m a.s.l. (Figure 9a and b).

The aggregated map including recharge areas of all discharge locations (Figure 9c) shows that within the probable recharge elevation range from 1200 m up to 2200 m a.s.l. recharge is biased towards convex-shaped landforms as opposed to concave-shaped landforms. The former are commonly characterised by a dense network of fractures. The prominent fractured ridge marking the western boundary of the DSGSD (Figure 9d, e and f) therefore seems to act as a preferential recharge zone, while groundwater exfiltration occurs preferentially in concave-shaped landforms.

## 330 4.7 Hydrogeological landslide control

Based on the maps of probable recharge areas it is evident that groundwater recharges over a large extent of the landslide's upslope catchment area. Whereas springs located on the landslide (e.g. L02 and L03) infer medium 3D flow distances in the order of 800 to 1400 m, those inferred for the well samples reach up to 3000 m (Figure 10). Consequently, pore pressure changes, triggering landslide activity, are controlled by a distal and a proximal forcing of groundwater recharge. Comparing

inferred 3D flow distances of all springs against their discharge elevation it becomes evident that the inferred flow distance increases with decreasing elevation (Figure 10). Moreover, springs characterised by longer flow distances indicate longer transit times reflected by smaller ITTP compared to springs with a close-by recharge area and higher ITTP. Simultaneously, this observation reminds of a similar pattern between increasing EC and decreasing discharge elevation (Figure 4a).

These observations suggest the existence of a homogeneous flow in a fractured aquifer of rather uniform hydrodynamical

properties (Figure 11). Recharge of this aquifer takes place between 1000 and 1650 m a.s.l. (Figure 9c). The upper boundary of this groundwater flow system coincides with elevations representing the ridge associated with the landslide suggesting a topographically controlled and almost slope-parallel flow system.

Another water transport mechanism and control of pore pressure changes within the landslide is located in the deeper subsurface below or at the lowest parts of the shallow coherent aquifer. Evidence for its existence is mainly based on the estimated recharge

areas of water sampled in the wells a few tens of meters below the surface. Inferred long flow distances indicate flow routes transporting water from the uppermost parts of the DSGSD towards the lower parts of the active landslide area. Considering the topographic properties within the DSGSD catchment, this transport mechanism requires slope-discordant flow paths which are at least twice as long as the flow distances estimated for the shallow flow system. These findings allowed to prepare a conceptual model of groundwater movement within the DSGSD in order to describe identified flow mechanisms along the

slope (Figure 11, Table 3).

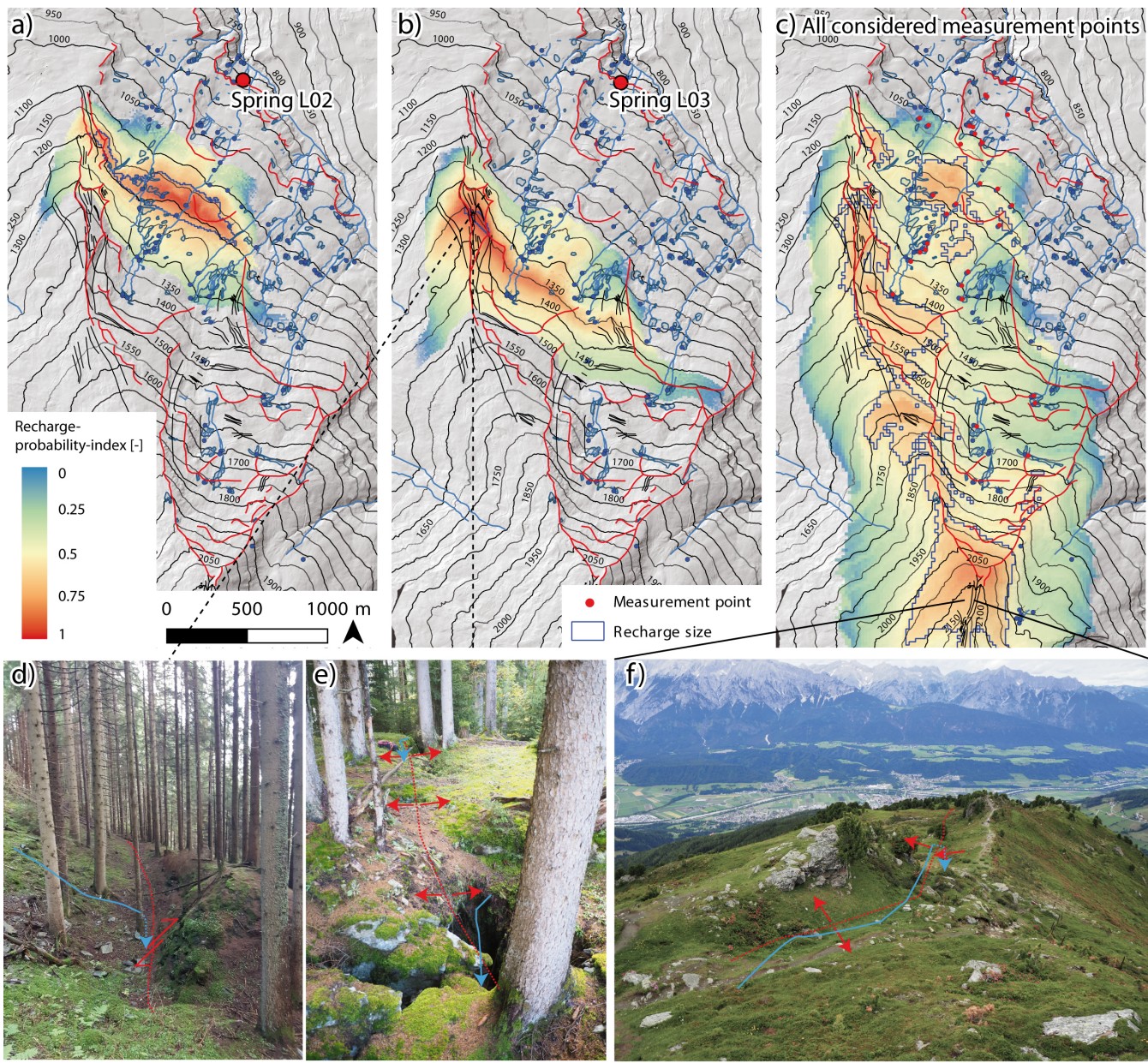

**Figure 9.** Maps of probable recharge. a) and b) show the recharge probability index for selected springs draining the active landslide and c) shows an aggregated map considering recharge probabilities for all spring, drainage and well samples. d) shows a picture of an upslope facing scarp assumed to represent a recharge zone (assumed flow paths indicated by blue arrow-lines) as well as e) open tensional fractures and f) a dry valley between a double-crested ridge (DTM source: Federal State of Tyrol).

**Table 3.** Summary of identified flow system and their properties.

|  | Proximal | Distal |
|---|---|---|
| Discharge elevation [m] a.s.l. | 800-1350 | 800 |
| Mean recharge elevation [m] a.s.l. | 1000 - 1650 | 1650 -2 200 |
| 3D flow distance [m] | 0- 1500 | Up to 3000 |
| $\delta^{18}$O [‰] | -12.0 to -11.5 | -13.1 to -12.5 |
| Description | Quick response, proximal recharge areas, shallow and short flow path | Slow response, distal recharge areas, long flow path |

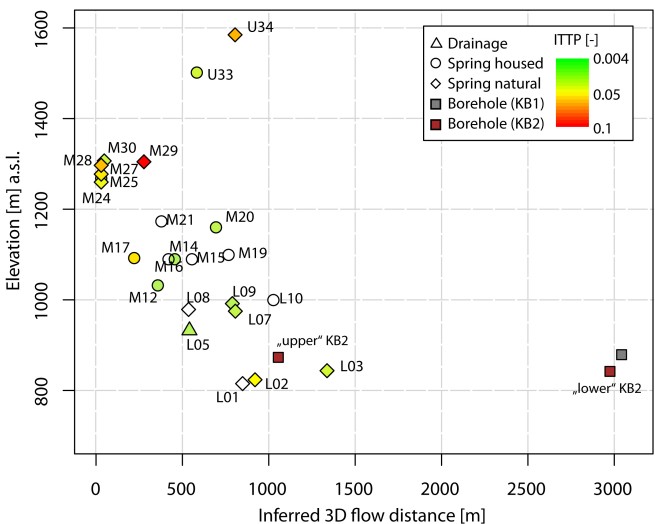

**Figure 10.** Relation between inferred 3D flow distance and discharge elevation. Colours represent calculated ITTP after Tetzlaff et al. (2009) where calculations were feasible. Sampling dates are in conformity with data points in Figure 7a.

## 5 Discussion

Hydrogeological monitoring combined with an automated geostatistical approach provides new insights into the hydrogeological drivers of the deep-seated Vögelsberg landslide. Monitoring of precipitation and groundwater enhanced the understanding of the hydrological control of the landslide. Determined stable isotopic compositions of groundwater and precipitation sam-

pled at different elevations allowed to determine probable recharge elevations. Combined with an index representing the 3D distances to discharge locations, recharge probability maps were reconstructed. Emphasis was placed on accurately estimating the local $\delta^{18}$O gradient of precipitation with respect to the sensitivity of the subsequent estimation of mean recharge elevation. It was found that the isotopic composition of locally sampled precipitation is in agreement with respective time series pro-

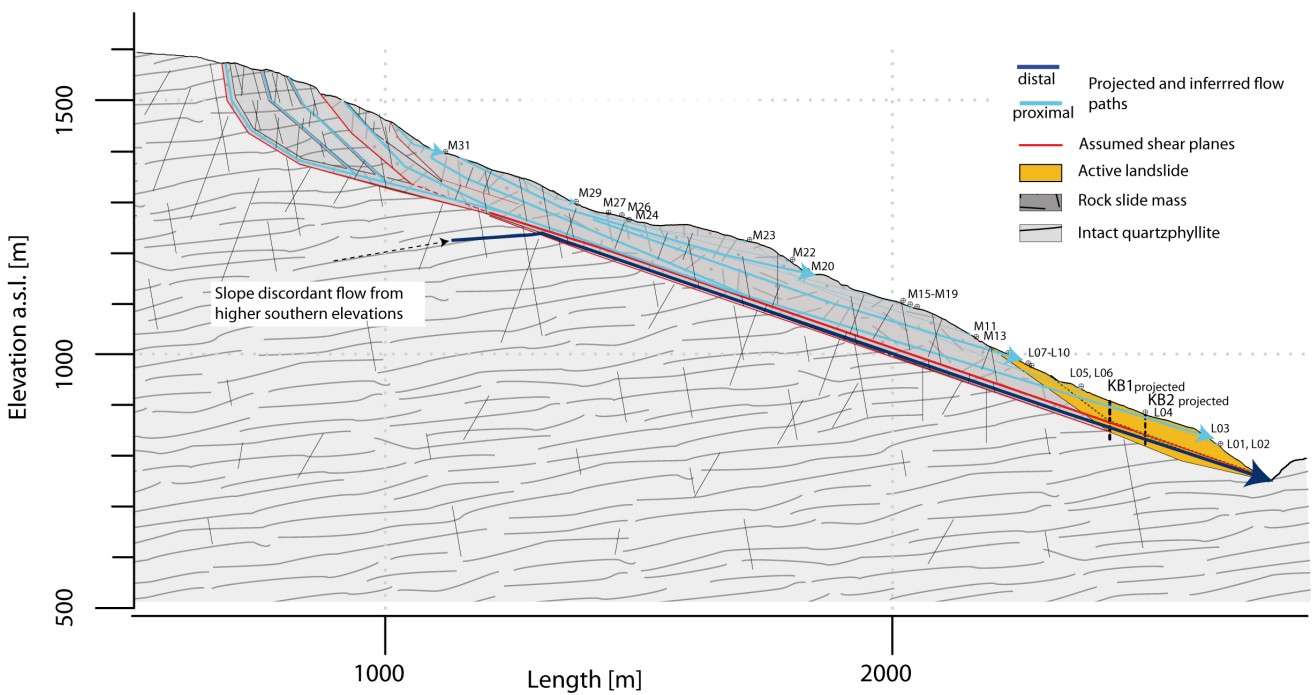

**Figure 11.** Conceptual hydrogeological model summarising dominant groundwater flow systems along a transect through the DSGSD and Vögelsberg landslide.

vided by ANIP. This strengthens the quality of the monitored data and the chosen monitoring approach and justifies the use of
the long-term ANIP data for a more robust derivation of probable recharge elevations. Identified altitude gradients of 0.14‰,
0.16‰ and 0.21‰ per 100 m are in the range of gradients in mountain regions around the globe (0.15–0.5‰ per 100 m; Clark
and Fritz (1997)) and in good agreement with the Austrian mean gradient of 0.16‰ per 100 m (Humer et al., 1995). Further-
more, these gradients are in good agreement with a value of 0.18‰ per 100 m identified by Madritsch and Millen (2007) in
a study approximately 5 km south of the Vögelsberg landslide. From this we propose an enlarged area of applicability for the
gradient-approach used in the present study. The confidence interval of the gradients should therefore be valid at least within
the area between the Vögelsberg study area, the Innsbruck and Patscherkofel ANIP stations and the study area of Madritsch
and Millen (2007).

The low temporal variability in the isotopic composition of the groundwater indicates significant attenuation of the seasonal
$\delta^{18}O$ signal by mixing and long transit times which are expressed by low ITTPs observed within the aquifer. These results
justify the usage of multi-annual $\delta^{18}O$ ratios of precipitation to derive altitude gradients in order to estimate the mean recharge
elevations of the springs. Although low ITTPs obtained in this study indicate transit times of up to 3.5 years (Tetzlaff et al.,
2009), our previous study (Pfeiffer et al., 2021) has shown that the landslide responds to hydro-meteorological events with a
delay of up to 50 days. This distinct disparity between response time and transit time indicates that the propagation of pore

pressure within the aquifer is multiple times faster than the subsurface flow itself. Although the calculation of ITTP was not feasible for the well samples (KB1 and KB2) due to too few multitemporal measurements, the constant $\delta^{18}O$ ratios between single measurement campaigns are interpreted as indicator for comparably long transit times.

Recharge probability maps indicate that groundwater recharges preferably at elevations between 1000 and 1650 m a.s.l. for springs emerging between 800 and 1350 m a.s.l. Water encountered in the wells recharges up to 2200 m a.s.l. The resulting recharge probability maps generally agree with field observations. Areas with high recharge probability are in agreement with mapped areas of geomorphological features supporting groundwater recharge, including open fractures, fissures, up-facing scarps, double crested ridges and a generally dry appearance of the humped terrain.

The hydrogeological characteristics of the study area and the low temporal $\delta^{18}O$ variability of the groundwater indicate a well-mixed aquifer. It is therefore assumed that the partially reworked and heavily fractured quartzphyllite supports isotropic flow directions. Such conditions are a prerequisite for using the proposed distance-weighted index for differentiating between high and low recharge probability at similar elevations. Therefore, we see a high potential for applying our approach to groundwater systems of similar conditions. In case of mainly anisotropic conditions (e.g. due to one or more prevailing geologic structures) the orientation of the anisotropy controlling the subsurface flow direction must be considered in the assessment of probable recharge areas.

Regarding the comparably higher recharge elevations of groundwater encountered in wells, flow paths following slope-discordant directions had to be introduced to explain groundwater transport, i.e. water flow along the basal shear zone representing a large-scale inhomogeneity at the lower boundary of the fractured and partially disaggregated quartzphyllite. A similar flow mechanism was proposed by Madritsch and Millen (2007) for a DSGSD on the opposite side of the valley. Intact quartzphyllite below the deformed layer is assumed to host almost impermeable hydrogeological conditions. Within the deformed layer slope-discordant groundwater flow is not excluded. Vallet et al. (2015) and Ronchetti et al. (2020) identified similar slope-respectively slide direction-discordant groundwater flow mechanisms with tracer tests at deep-seated landslides.

The concordance of discharge time-series with continuous time-series of groundwater level, landslide displacement rate and precipitation time-series indicate that the chosen monitoring interval is sufficient to capture the hydrodynamic behaviour of the respective aquifer in time (Figure 6). Furthermore, correlations of discharge and landslide velocity indicate that the aquifer controls the landslide's activity. Evidenced by temporal dynamics in the aquifer (spring and piezometer), the three landslide triggering hydro-meteorological events observed and analysed by Pfeiffer et al. (2021) between 2016 and 2019 are extended by three additional landslide acceleration events (summer 2019, late winter/spring 2020 and late summer/autumn 2020) (Figure 6). Differences in the temporal variability of discharge among springs in the study area show an elevation-staggered behaviour. Discharge of springs at lower elevations increases earlier than the discharge at higher elevation springs. This behaviour was characteristic for the snowmelt season in 2019 where for the same timespan a fast landslide response to a distinct snow melt event at elevations below 1700 m a.s.l. was observed (Pfeiffer et al., 2021). Results of the present study suggest that this landslide acceleration event was triggered by recharge of the shallow parts of the aquifer right above the active landslide. Snowmelt started earlier at lower elevations (approx. 1000 m a.sl.) and successively moved to elevations of up to 1600 m a.s.l.. Infiltration due to snowmelt is assumed to be faster than for rainfall and a concomitant pore-pressure increase induces an almost immediate

acceleration of the landslide movement.

In addition to this proximal, shallow and topographically controlled flow system, recharge elevations of the deeper groundwater indicate a more distal deep-seated flow system which transports water originating in the uppermost areas of the DSGSD catchment. Rainfall-induced landslide acceleration as observed by Pfeiffer et al. (2021) in 2016 and 2017 is assumed to be the response to both, recharge of the proximal flow system as well as recharge and pore pressure rise caused by the distal flow system. The longer response time of 24 to 50 days estimated for these events therefore is the consequence of the longer distance

of up to 3000 m between recharge area and landslide. With the available data we could not identify if there is distinct mixing between the deeper distal flow and the shallow proximal flow system. Nevertheless, we do not exclude a vertical exchange between shallow and deeper water. It is likely that the distal flow system occurs below the proximal flow system and along the lower boundary of the deformed rock mass.

Similar hydrogeological mechanisms were described by Guglielmi et al. (2002) at the La Clapière and Séchilienne landslides,

where also a shallow and a deep groundwater flow system were proposed. Comparable to our study, Vallet et al. (2015) interpreted potential recharge elevations for the Séchilienne landslide. They used the estimated recharge elevation and its intersection along the slope line or on a slope-discordant line reaching higher topographic elevations to distinguish between topographic controlled and structural controlled subsurface flow paths. The maps of probable recharge areas for the Vögelsberg landslide show comparable results. Thus, our method provides objective information for planning mitigation measures to

drain unstable slopes. On the other hand, future drainage systems aiming at reducing groundwater recharge could modify the $\delta^{18}$O composition of springs, and hence, the recharge probability maps. In this way, the impact of the mitigation strategy could be evaluated and subsequently improved.

## 6 Conclusions

This study presents a new approach for assessing groundwater recharge areas to improve the understanding of the hydrogeology

of slopes. The highly automated geo-statistical approach yields recharge probability maps based on stable isotope data and a DEM. It was applied to the Vögelsberg landslide, a currently active slab of a DSGSD in the Watten valley (Tyrol, Austria). Local $\delta^{18}$O-altitude gradients of precipitation are in agreement with gradients derived from long-term measurements at stations of the ANIP network. The established local $\delta^{18}$O-altitude gradient allowed to estimate the mean recharge elevation of groundwater sampled at springs and in two wells. The recharge areas were then further constrained based on 3D distances to the spring

locations, computed with the help of a DEM.

The resulting recharge probability maps suggest a dual hydrogeological control: a proximal control provided by a shallow aquifer recharging between 1000 to 1650 m a.s.l. and distal flow of groundwater originating in the uppermost areas of the DSGSD. Both, the distal and the proximal flow mechanisms are compared and in accordance with the previously identified hydro-meteorological triggering of landslide acceleration events at this site (Pfeiffer et al., 2021).

The study illustrates the strength of stable water isotopes as a natural tracer to identify and constrain groundwater flow paths.

These data helped to fully understand the hydrogeological controls of the Vögelsberg landslide. Furthermore, the recharge probability maps provide valuable information for mitigation measures (e.g. drainage systems).

*Data availability.* The hydrogeological data that support the findings of this study are openly available in the Zenodo repository at: http://doi.org/10.5281/z

*Author contributions.* JP, TZ and JS developed the research concept, acquired necessary field data and carried out the analysis. JP drafted
the manuscript. TZ, JS, TB, MR and CS reviewed the manuscript and all authors contributed to the editing and revision process.

*Competing interests.* The authors declare no conflicts of interest.

*Acknowledgements.* The present study was conducted within the OPERANDUM project. This project has received funding from the European Union's Horizon 2020 research and innovation programme under grant agreement No. 776848. We thank the Federal State of Tyrol ("Abteilung Geoinformation" and "Landesgeologie") and the Austrian Service for Torrent and Avalanche Control for providing data and
valuable exchange of information. We thank Tim Phillip for analysing the water samples and Sebastian Gehring for assistance in the field.

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
