# Peer review of "Spatial assessment of probable recharge areas - Investigating the hydrogeological controls of an active deep-seated gravitational slope deformation"

_Natural Hazards and Earth System Sciences, 2021_

## Referee Comment (RC1)

Review of the article

Spatial assessment of probable recharge areas - Investigating the hydrogeological controls of an active deep-seated gravitational slope deformation

J. Pfeiffer et al.

The subject is of major interest in mountainous context where it is often difficult to make a water balance in this kind of environment. There are many reasons for this: difficulties in estimating the recharge area, difficulties in estimating the outflows, because they are frequently masked by quaternary formations or rivers. To make a water balance of an aquifer allows to better manage the water resource, which in mountainous environment is fundamental because these are often the only water resource of small communities.

This paper proposes a methodology for estimating recharge areas in particular environments that are gravity instabilities. In these environments, estimating the search area is a real issue, because it is recognized that water is an aggravating factor in the displacement of these slopes. Any identification of water inflow is therefore essential to be able to set up remediation systems (example of drainage cited by the authors).

The method developed in this study is original and seems to give promising results. Unfortunately, there are many inaccuracies in both form and substance that taint these results.

On the form:

The figures are not all good quality and need to be revised

**Figure 1**

the DSGSD's right of way should be better marked

Are all the sources represented perennial? What guided the choice to follow certain sources rather than others?

On the location of the site, it would have been nice to indicate a known city (for those who do not know the Austrian Alps)

**Figure 2**

The identification of the sampling points is not intuitive for someone who is not familiar with the area. Moreover, it is written relatively small. This identification appears on figures 4, 5, 7 and 10 and really weighs down these figures, which does not facilitate reading. Maybe we should think about another simplified annotation.

What is the difference between "housed spring" and "natural spring"? Is the water from the "housed spring" captured subsurface water?  Can you clarify? Is it necessary to differentiate between the two types of springs?

Does Figure 2c also include the rain sampler and snow sampler photos? Same for figure 2d? It probably does, but you can improve the readability of the figure by putting frames around the "photos that make up Figure 2C and the 4 photos in Figure 2d.

**Figure 4**

This figure could be improved by changing the names of the sampling points. It is not possible to differentiate the physico-chemical parameters of the water taken from the boreholes at different depths. There are two boreholes and three different symbols (red square, black square, and grey square). Is it a way to identify the samples taken at different hydrological periods in these wells? This is not intuitive and yet it is important information in terms of identifying water masses.

Could you specify the date of sampling and indicate the hydrological season (low water level, high water level)

**Figure 5**

This temporal figure is not very readable for some curves. The choice of colours for some curves should be reviewed. As there is not much variation for the electrical conductivity for temperature, you could add a table with statistical data would have been enough to describe the behaviour of this parameter (min, max values, standard deviation).

Concerning the O18 isotopes, why not have represented the $\delta$ H2 as a function of the $\delta$ O18 using the meteoric line even if there is no local meteoric line. This representation gives information on the origin of the water and makes it possible to differentiate infiltration water according to the seasons. This would have been a plus in your demonstration.

**Figure 7**

It can also be improved in the choice of colours for the flows.

Why not correlate these data with the precipitation data?

**Figure 10**

What colour code did you use to identify the points?

Is it to discriminate them from the colour code chosen for the ITTP? If so, why are there dark dots that do not appear in the ITTP scale?

Did you represent all the points? And if not, can you explain why?

Could you specify the date of sampling and indicate the hydrological season (low water level, high water level)?

**Calling figures in the text: it is a bit anarchic**

In paragraph 2 describing the Vögelsberg landslide, only figure 1a is called. It is not until paragraph 4.1 Hydrogeological characterization that figures 1 b, c and d are mentioned. This is very surprising. Paragraph 2 states the knowledge of the slide so if these measurements represented on figures 1b to d are new they must be integrated in another figure.

Figure 2b called before figure 2, figure 2c called before figure 2a. It seems to me that there must be an order to respect in the calling of the figures which is not the case on almost all the figures in this article.

Figure 3b is called before 3a and I stop my inventory there. I leave it to the editor to let you know if the order of calling the figures is important in the articles of this journal.

**Paragraph 2**

I am surprised to see no hydrogeological analysis in this paragraph. There are however things that exist since you have the inventory of springs. Has this hydrogeological description never been done? It is surprising because working directly on the recharge without having an idea of the hydrological context.

**Paragraph 3.1**

The whole sampling protocol is rather confusing and should be rewritten for a better understanding.

For example, I did not understand the measurements made in the drillings. Could you specify at what depth the measurements were made? This is important at least for borehole KB2 because you have an electrical conductivity that varies significantly over 18m which is not the case for borehole KB1 which raises questions about the origin of the water masses in these boreholes.

**Paragraph 3.2**

No need to quote again Tetzlaff et all (2009) on line144 because already quoted in the same paragraph on line 142

**Paragraph 4.2**

Why not compare the responses of springs in terms of flow variation with precipitation? This is usually done in classical hydrological studies. This gives indications on the response time of the aquifer but also on the transit times.

**On the content**

The introduction poses the problem well except that the authors remain too much in a context of hydrogeological investigations in unstable massifs.

Concerning the use of isotopes for hydrogeological studies, there are many publications that are not necessarily related to a gravity context, which describe very well the interest of oxygen isotopes in the characterization of the origin of water masses, the seasonal effect of infiltration and on the infiltration altitudes.

there are references more accessible to the community at large than Moser and Rauert, if you want to leave this ref add one more accessible

page 45: missing are geophysical methods that are increasingly used to characterize variations in groundwater flow and storage in gravity contexts. These are non-invasive methods that are being used successfully. I think we need to add more references on this topic.

**Results**

The authors describe the fractured medium as an equivalent porous place, which is consistent with their value of transit time of a little more than 3.5 years. A brief theoretical calculation confirms this assumption with a theoretical hydraulic conductivity of the order of 10-4 m/s (displacement of a 50m)

The coupled analysis of conductivity, temperature and ITTs allows to propose a conceptual model of the flows in DSGSD. Here, conductivity is used as a tracer of transit times. Using only electrical conductivity as a transit time can lead to misunderstanding. The authors explain the increase in electrical conductivity because of increased upstream-downstream mineralization due to longer water-rock interaction time due to longer upstream-downstream transit times.

The conductivity of the water varies from 80µS/cm to about 600µS/cm. This corresponds approximately to waters with a TDS (total dissolved solids) between 52mg/l and 450 mg/l.

In this part, we are missing some important information to understand these variations of electrical conductivity. The electrical conductivity reflects the mineralization of the water. If we are dealing only with silicate minerals, the water coming from this environment is not very mineralized, whatever the time of water-rock interaction. This is because the kinetics of silicate minerals are very weak under surface conditions. Therefore, 450mg/l of TDS cannot be explained solely by the dissolution of these minerals.

There are two possibilities

- Either the waters flow only on silicates (less mineralized waters), or the waters have a carbonate signature (due to the influence of marbles? Or other minerals more soluble ?).

A final explanation may be a mixture of water between these two poles (silicate and carbonate). One thing is sure, electrical conductivity is a tracer of mineralization and not of transit time.

This is what Hilbert (2016) writes . « The groundwater quality depends on the aquifer lithology, and is therefore very variable with mostly alkaline pH-values and higher mineralisation in carbonate units, while slightly acidic groundwater and low mineralisation are typical in the fractured parts of the mountain range (Austrian Geological Survey 2004)

The authors describe the geology as part of the Quartzphylite complex, whose minerals are mostly silicate with marble intercalations. This description is not sufficient to understand the true electrical conductivity

Peter TROPPER* & Andreas PIBER (2012) Geothermobarometry of quartzphyllites, orthogneisses and greenschists of the Austroalpine basement nappes in the northern Zillertal (Innsbruck Quartzphyllite Complex, Kellerjochgneiss, Wildschönau Schists; Tyrol, Eastern Alps) describe the mineralogy more precisely. This article could help the authors in their geological characterization.

In the absence of a chemical characterization of the water bodies, I am not convinced by their argument that the further downstream the source is, the longer the distance travelled, the longer the transit time. If this is the overall trend, how can we explain the points 093_001 (high altitude, inferred distance of about 1000 and conductivity about 100µS/cm? 103-001 high altitude, inferred distance of approximately 500 and conductivity approximately 200µS/cm? and points 047-001 and 048_001 which are at the same altitude with an inferred distance of double and a electrical conductivity around 300µS/cm.

Sources with high EC values but low flow path lengths (e.g. 064-001, 062-001 and 018-001) are influenced by longer residence times (e.g. low ITTP) resulting in comparatively slower flow as is the case for sources with lower EC values at the same altitude (Figure 10b): this may simply be a mineralogical effect.

I think there is a bias in the interpretation

While there is indeed a correlation between infiltration distance and transit time, there is no correlation between transit time and mineralization, especially in these complex environments where all it takes is one soluble mineral to significantly mark the mineralization of the water

The arguments of Kilberg (2016) on electrical conductivity should be considered again

"Specific electrical conductivity (EC) is a key parameter for quantifying total water mineralization (the physical background is given for example by Matsubayashi et al. 1993). EC is primarily controlled by dissolution processes within the aquifer; therefore, the parameter can be used as an indicator for various aquifer lithologies. Due to the variable solubility of the mineral phases, carbonate and even more evaporite aquifers are characterized by highly mineralized groundwater, whereas silicate lithologies contain groundwater with comparatively lower EC values (Kilchmann et al. 2004)".

Have you tried to analyze the low and high-water level representative points separately? In low water level, the conductivity signal represents the water/rock interaction signal, whereas in high water level , it is rather the infiltration signal. This may shed new light on your interpretation.

One last comment:

I disagree with your argument about temperature increasing with decreasing elevation as a marker of transit time. It would be necessary to specify the dates of sampling on figure 4, it seems to me that this is not indicated and can be seen with the precipitation. When we analyse the temperature time series, we can see the seasonal variations. The minima seem to be around 5°C except perhaps for the lowest sources. Obviously, the peaks are shifted in time, the graph is not of very good quality. It is worth reanalysing these data.

In conclusion,

This is an interesting article that develops a methodology for estimating recharge as a function of water infiltration altitude using O18 isotopes and a numerical field model that allows us to constrain these recharge areas.

I think that the flow part still lacks data, especially on the different water masses. The only measurement of the electrical conductivity is not sufficient to constrain these flows and cannot in any case be used as a tracer of transit time in complex geological environments.

A suggestion for your next study

I would start by mapping the water outlets (temperature, pH (the pH would allow to isolate the flows on the silicates) and electrical conductivity) that I would place on a geological support (look at where these marble banks are in relation to the outlets).

Then I would select springs which are differentiated by these parameters and make high frequency measurements of electrical conductivity, water height (or flow) on a transect.

I would also equip the boreholes with this type of probe (does the bottom of the boreholes reach the fracture surface? If so, it's good because these surfaces are major drains of flows.

Nevertheless, I find your approach to determine the recharge zones really very good, your hydro study a little less so

---

## Author Comment (AC1)

**Response to the Reviewers' comments**

(original comment in italics and response in green)

Reviewer #1 (Catherine Bertrand)

https://nhess.copernicus.org/preprints/nhess-2021-388/#RC1, 2022

*Review of the article*
*Spatial assessment of probable recharge areas - Investigating the hydrogeological controls of an active deep-seated gravitational slope deformation J. Pfeiffer et al.*

*The subject is of major interest in mountainous context where it is often difficult to make a water balance in this kind of environment. There are many reasons for this: difficulties in estimating the recharge area, difficulties in estimating the outflows, because they are frequently masked by quaternary formations or rivers. To make a water balance of an aquifer allows to better manage the water resource, which in mountainous environment is fundamental because these are often the only water resource of small communities.*

*This paper proposes a methodology for estimating recharge areas in particular environments that are gravity instabilities. In these environments, estimating the search area is a real issue, because it is recognized that water is an aggravating factor in the displacement of these slopes. Any identification of water inflow is therefore essential to be able to set up remediation systems (example of drainage cited by the authors).*

*The method developed in this study is original and seems to give promising results. Unfortunately, there are many inaccuracies in both form and substance that taint these results.*

*On the form:*
*The figures are not all good quality and need to be revised*
***Figure 1***
*the DSGSD's right of way should be better marked*
*Are all the sources represented perennial? What guided the choice to follow certain sources rather than others? On the location of the site, it would have been nice to indicate a known city (for those who do not know the Austrian Alps)*

We are thankful for these suggestions and changed Figure 1 accordingly. The revised figure better shows the DSGSD extent, indicates if a spring is perennial, episodic or if its behaviour is unknown. Identification of episodic springs requires repeated field visits which was not done for all springs shown on the figure. For springs that were frequently monitored we classified them into perennial and episodic ones. Springs that never fell dry in the monitoring period are therefore highlighted as perennial springs, while springs that fell dry are marked as episodic springs. Springs that were visited once (to create the spring inventory and not selected for repeated measurements) are marked as "unknown".
By adding cities (e.g. Innsbruck, Vienna) to the overview map the location of the landslide should be more comprehensible.

Selection of springs was done because of their assumed proximity to the landslide's area of influence. We added a sentence explaining the selection approach to the manuscript (Line 161-165).

".... Hydrogeological monitoring campaigns were carried out from October 2018 until June 2020. Based on the hydrogeological inventory provided by the Federal State of Tyrol (Figure 1a), 35 measurements points fulfil the demands to be part of a temporally condensed measurement setup. Selection of measurement points was done based on the following criteria: measurement points are accessible and permitted to be accessed by the owner during the monitoring period, natural water outlets are

effectively measurable without disturbance of the surrounding environment, and measurement points intersect with the assumed area of potential landslide influence. ..."

**Figure 2**
*The identification of the sampling points is not intuitive for someone who is not familiar with the area. Moreover, it is written relatively small. This identification appears on figures 4, 5, 7 and 10 and really weighs down these figures, which does not facilitate reading. Maybe we should think about another simplified annotation.*

We welcome this suggestion and introduced a spatially and elevation-dependent annotation system. Additionally, a description of the annotation and a labelling of the measurement points were added to the manuscript:

L165- L170

"....The assumed area of potential landslide influence covers the lowest and highest part of the DSGSD and is grouped into three elevation bands (Figure 2b). Measurement point designation is based on the discharge elevation and prefixed acronym indicating the elevation band. Measurement points L01 – L10 intersect with the sections of the lower slope and the area of the active landslide (<1000 m a.s.l.). Measurement points M11 – M31 intersect with the middle slope section (1000 – 1500 m a.s.l.) and U32-U35 with the upper slope (>1500 m a.s.l.).

*What is the difference between "housed spring" and "natural spring"? Is the water from the "housed spring" captured subsurface water? Can you clarify? Is it necessary to differentiate between the two types of springs? Does Figure 2c also include the rain sampler and snow sampler photos? Same for figure 2d? It probably does, but you can improve the readability of the figure by putting frames around the "photos that make up Figure 2C and the 4 photos in Figure 2d.*

By adding two sentences in Line 173 we aimed to clarify the differences between housed and natural springs. "…Housed springs are structurally supported groundwater outlets and relevant for the residents' water supply. Natural springs are mostly diffuse zones of groundwater exfiltration…"
In our opinion this differentiation contains fundamental information that supports the interpretation of these two types of springs. For the natural springs which commonly have a diffuse appearance and low discharge we expect a bias in the measurements due to potentially longer water-atmosphere interactions compared to housed springs with commonly higher and more concentrated discharge.

As suggested, we improved the alignment of the subfigures to enhance the structure and readability of Figure 2.

**Figure 4**
*This figure could be improved by changing the names of the sampling points. It is not possible to differentiate the physico-chemical parameters of the water taken from the boreholes at different depths. There are two boreholes and three different symbols (red square, black square, and grey square). Is it a way to identify the samples taken at different hydrological periods in these wells? This is not intuitive and yet it is important information in terms of identifying water masses.*
*Could you specify the date of sampling and indicate the hydrological season (low water level, high water level)*

We appreciate the careful examination of the figure and made changes as suggested. Labels of data points were revised to enhance readability and ensure consistency throughout the manuscript. Colours of boreholes were revised and a new sub-figure (**figure 4d**) was introduced. Here we added a more

detailed diagram showing the course of EC and $\delta^{18}$O along the boreholes in response to criticism raised by the reviewer. Furthermore, we moved the former Figure 4d ($\delta^{18}$O precipitation time series) to the restructured **Figure 7b.**

As suggested we indicated the date of borehole sampling within the piezometric time-series shown in Figure 6 in order to ensure comprehensibility of sampling data and its hydrological setting (e.g. water level, springs discharge, precipitation).

***Figure 5*** (Figure 7 after restructuring)
*This temporal figure is not very readable for some curves. The choice of colours for some curves should be reviewed. As there is not much variation for the electrical conductivity for temperature, you could add a table with statistical data would have been enough to describe the behaviour of this parameter (min, max values, standard deviation).*
*Concerning the O18 isotopes, why not have represented the δH2 as a function of the δO18 using the meteoric line even if there is no local meteoric line. This representation gives information on the origin of the water and makes it possible to differentiate infiltration water according to the seasons. This would have been a plus in your demonstration.*

We agree about the poor readability and revised **Figure 5 (=Figure 7 after figure restructuring)** which contains only the $\delta^{18}$O time-series of springs, whereas statistical data (mean, standard deviation, and number of measurements) of the other physico-chemical parameters were transferred to a new table. Additionally, we added a sub-figure (**figure 7c**) showing the $\delta^{2}$H as a function of $\delta^{18}$O for all isotope samples (precipitation and groundwater) presented in this study. Overlain by groundwater $\delta^{2}$H/$\delta^{18}$O, the contribution of different seasons to the groundwater samples is now made transparent.

We added following text to section 4.4 L355: "….Considering both, δ18O and δ2H values, the groundwater samples are aligned along the precipitation data and therefore agree with the local meteoric water line. All groundwater samples are evenly surrounded by higher and lower δ18O and δ 2H precipitation values indicating a balanced seasonal recharge (Figure 7c)…."

***Figure 7*** (Figure 6 after restructuring)
*It can also be improved in the choice of colours for the flows.*
*Why not correlate these data with the precipitation data?*

We revised the choice of colour for discharge time series that now follow a spectral colour range in concordance with the spring's elevation which is now additionally mentioned in the figure.
We refrained from a detailed correlation of the groundwater time-series with precipitation since this was already done in a previous study (Pfeiffer et al. 2021). Therein spatially distributed simulated water availability (rainfall, snowmelt corrected for evapotranspiration) was cross-correlated with landslide displacement time series. Nevertheless, we agree that adding a precipitation time-series to the figure improves completeness and quality. Therefore, we introduced a precipitation time-series from the Patscherkofel station.

***Figure 10***
*What colour code did you use to identify the points?*
*Is it to discriminate them from the colour code chosen for the ITTP? If so, why are there dark dots that do not appear in the ITTP scale?*
*Did you represent all the points? And if not, can you explain why?*
*Could you specify the date of sampling and indicate the hydrological season (low water level, high water level)?*

The colour code refers to the ITTP. Former black points were changed to non-coloured points and represent points were the ITTP calculation was not feasible due to an insufficient number of multitemporal measurements. Dates of sampling go along with the respective period of consecutive measurements. They can easily be retrieved from the $\delta^{18}O$ time series shown in Figure 7a. For clarification we added the following sentence in the caption: "Sampling dates are in conformity with data points in Figure 7a."

**Calling figures in the text: it is a bit anarchic**

*In paragraph 2 describing the Vögelsberg landslide, only figure 1a is called. It is not until paragraph 4.1 Hydrogeological characterization that figures 1 b, c and d are mentioned. This is very surprising. Paragraph 2 states the knowledge of the slide so if these measurements represented on figures 1b to d are new they must be integrated in another figure.*

*Figure 2b called before figure 2, figure 2c called before figure 2a. It seems to me that there must be an order to respect in the calling of the figures which is not the case on almost all the figures in this article. Figure 3b is called before 3a and I stop my inventory there. I leave it to the editor to let you know if the order of calling the figures is important in the articles of this journal.*

We agree on the potential to increase the readability of the manuscript by re-structuring the calling, label and position of figures. As a side effect of revising and reorganising the structure of the manuscript (e.g. moving former section 4.1 "Hydrogeological characterization" to section 2, The Vögelsberg landslide), simultaneously the calling of figures in the text was structured in a more chronological way. We now also call the sub figures (a,b,c,… ) in an alphabetic and chronological order. We restructured the figures within the manuscript, to achieve a chronological order of calling the main figure within the text. Since we placed great emphasis on both, a hierarchically structured presentation of results throughout the text and a topic-supported structure of individual figures, some subfigure calls occur later at appropriate text sections.

**Paragraph 2**

*I am surprised to see no hydrogeological analysis in this paragraph. There are however things that exist since you have the inventory of springs. Has this hydrogeological description never been done? It is surprising because working directly on the recharge without having an idea of the hydrological context.*

As mentioned in 3.1 the spring inventory was adapted from the "Landesgeologie" (Federal State of Tyol). A detailed hydrogeological description has not been done so far and was partly the goal of present study. We moved the former section 4.1 (describing the hydrogeological characteristics of the Vögelsberg DSGSD) to the study area description (section 2).

**Paragraph 3.1**

*The whole sampling protocol is rather confusing and should be rewritten for a better understanding. For example, I did not understand the measurements made in the drillings. Could you specify at what depth the measurements were made? This is important at least for borehole KB2 because you have an electrical conductivity that varies significantly over 18m which is not the case for borehole KB1 which raises questions about the origin of the water masses in these boreholes.*

Thanks to the reviewer's suggestion we put effort into enhancing the description of field work and sampling protocol. We restructured individual paragraphs and added detailed description of the measurement strategy (L161-170). Furthermore we added a definition of housed and natural springs. For clarification of the measurements done in the boreholes we added the following sentence in L180: "…Well measurements and sampling was done at constant intervals from the piezometric height towards the bottom of the well…"

**Paragraph 3.2**

*No need to quote again Tetzlaff et all (2009) on line144 because already quoted in the same paragraph on line 142*

Changed as suggested

*Paragraph 4.2*
*Why not compare the responses of springs in terms of flow variation with precipitation? This is usually done in classical hydrological studies. This gives indications on the response time of the aquifer but also on the transit times.*

We did not compare the spring response to precipitation mainly because in the former study (Pfeiffer et al., 2021) we already compared hydro-meteorological time-series with landslide displacement time series. However, we agree with the reviewer and now include a precipitation time series from the ANIP Patscherkofel station. At appropriate sections (e.g. section 4.2 and section 5) we additionally present and discuss the added value of the precipitation time series in the revised manuscript. E.g. in L 318: "… The accelerated landslide movements during the period of higher water levels and increased spring discharge in early 2019 were the response to intensive snowmelt, as shown by a physically-based snow model (Pfeiffer et al., 2021). The aquifer response to this and subsequent snowmelt and summer precipitation events is indicated by the comparison of respective groundwater time series with precipitation time series at the close-by Patscherkofel station (Figure 6)…"

*On the content*
*The introduction poses the problem well except that the authors remain too much in a context of hydrogeological investigations in unstable massifs.*
*Concerning the use of isotopes for hydrogeological studies, there are many publications that are not necessarily related to a gravity context, which describe very well the interest of oxygen isotopes in the characterization of the origin of water masses, the seasonal effect of infiltration and on the infiltration altitudes.*

We reviewed existing literature and added references that are not solely related to a gravity context to support and enhance the stable isotope data in a wider context. Furthermore, we adapted and more precisely extended the text by additionally presenting not-landslide related studies that describe well the functionality of stable isotopes for characterising the origin of water masses (e.g. Schmieder et al. 2016), demonstrating the seasonal effect of infiltration (e.g. Jasechko et al. 2014) and assessing recharge elevations (e.g. Blasch and Bryson 2007).

*there are references more accessible to the community at large than Moser and Rauert, if you want to leave this ref add one more accessible*

We added a second reference about the isotope elevation gradient (e.g. Blasch et al. 2007).

*page 45: missing are geophysical methods that are increasingly used to characterize variations in groundwater flow and storage in gravity contexts. These are non-invasive methods that are being used successfully. I think we need to add more references on this topic.*

We added selected references of geophysical methods for non-invasive characterisation of groundwater conditions and dynamics: Jomard et al. 2007, Siemon et al. 2009, Chalikakis et al. 2011, Zieher et al. 2017, Lajaunie et al. 2019.

*Results*
*The authors describe the fractured medium as an equivalent porous place, which is consistent with their value of transit time of a little more than 3.5 years. A brief theoretical calculation confirms this assumption with a theoretical hydraulic conductivity of the order of 10-4 m/s (displacement of a 50m) The coupled analysis of conductivity, temperature and ITTs allows to propose a conceptual model of the flows in DSGSD. Here, conductivity is used as a tracer of transit times. Using only electrical conductivity as a transit time can lead to misunderstanding. The authors explain the increase in*

*electrical conductivity because of increased upstream-downstream mineralization due to longer water-rock interaction time due to longer upstream-downstream transit times.*

*The conductivity of the water varies from 80µS/cm to about 600µS/cm. This corresponds approximately to waters with a TDS (total dissolved solids) between 52mg/l and 450 mg/l.*

*In this part, we are missing some important information to understand these variations of electrical conductivity. The electrical conductivity reflects the mineralization of the water. If we are dealing only with silicate minerals, the water coming from this environment is not very mineralized, whatever the time of water-rock interaction. This is because the kinetics of silicate minerals are very weak under surface conditions. Therefore, 450mg/l of TDS cannot be explained solely by the dissolution of these minerals.*

*There are two possibilities*

*- Either the waters flow only on silicates (less mineralized waters), or the waters have a carbonate signature (due to the influence of marbles? Or other minerals more soluble ?).*

*A final explanation may be a mixture of water between these two poles (silicate and carbonate). One thing is sure, electrical conductivity is a tracer of mineralization and not of transit time.*

*This is what Hilbert (2016) writes . « The groundwater quality depends on the aquifer lithology, and is therefore very variable with mostly alkaline pH-values and higher mineralisation in carbonate units, while slightly acidic groundwater and low mineralisation are typical in the fractured parts of the mountain range (Austrian Geological Survey 2004)*

*The authors describe the geology as part of the Quartzphylite complex, whose minerals are mostly silicate with marble intercalations. This description is not sufficient to understand the true electrical conductivity*

*Peter TROPPER\* & Andreas PIBER (2012) Geothermobarometry of quartzphyllites, orthogneisses and greenschists of the Austroalpine basement nappes in the northern Zillertal (Innsbruck Quartzphyllite Complex, Kellerjochgneiss, Wildschönau Schists; Tyrol, Eastern Alps) describe the mineralogy more precisely. This article could help the authors in their geological characterization.*

*In the absence of a chemical characterization of the water bodies, I am not convinced by their argument that the further downstream the source is, the longer the distance travelled, the longer the transit time.*

*If this is the overall trend, how can we explain the points 093_001 (high altitude, inferred distance of about 1000 and conductivity about 100µS/cm? 103-001 high altitude, inferred distance of approximately 500 and conductivity approximately 200µS/cm? and points 047-001 and 048_001 which are at the same altitude with an inferred distance of double and a electrical conductivity around 300µS/cm.*

*Sources with high EC values but low flow path lengths (e.g. 064-001, 062-001 and 018-001) are influenced by longer residence times (e.g. low ITTP) resulting in comparatively slower flow as is the case for sources with lower EC values at the same altitude (Figure 10b): this may simply be a mineralogical effect. I think there is a bias in the interpretation*

*While there is indeed a correlation between infiltration distance and transit time, there is no correlation between transit time and mineralization, especially in these complex environments where all it takes is one soluble mineral to significantly mark the mineralization of the water*

*The arguments of Kilberg (2016) on electrical conductivity should be considered again*

*"Specific electrical conductivity (EC) is a key parameter for quantifying total water mineralization (the physical background is given for example by Matsubayashi et al. 1993). EC is primarily controlled by dissolution processes within the aquifer; therefore, the parameter can be used as an indicator for various aquifer lithologies. Due to the variable solubility of the mineral phases, carbonate and even more evaporite aquifers are characterized by highly mineralized groundwater, whereas silicate lithologies contain groundwater with comparatively lower EC values (Kilchmann et al. 2004)".*

*Have you tried to analyze the low and high-water level representative points separately? In low water level, the conductivity signal represents the water/rock interaction signal, whereas in high water level , it is rather the infiltration signal. This may shed new light on your interpretation.*

*One last comment:*

*I disagree with your argument about temperature increasing with decreasing elevation as a marker of transit time. It would be necessary to specify the dates of sampling on figure 4, it seems to me that this*

*is not indicated and can be seen with the precipitation. When we analyse the temperature time series, we can see the seasonal variations. The minima seem to be around 5°C except perhaps for the lowest sources. Obviously, the peaks are shifted in time, the graph is not of very good quality. It is worth reanalysing these data.*

We welcome the detailed explanations, professional criticism and suggestions for present and future studies. Furthermore, we agree with the reviewer's comments on the controls and process of water mineralisation and extensively revised our statements in the manuscript. The former idea was to discuss/evaluate the isotope-tracer-based results of recharge areas, inferred flow distances and flow paths by another (independent) variable (e.g. electrical conductivity). Mainly because we observed comparable relationships between electrical conductivity and discharge elevation we simply put similar observations between isotope-based-flow path length and discharge elevation together with the intention to strengthen the interpretations of the hydrogeological model. Thanks to the precise and accurate explanations of the reviewer we improved our understanding of the physics behind the utilised parameter and carefully revised our statements on the proposed interpretation based on observed relationship between EC/T and flow distance. The revised text describing the hydrogeological landslide control now solely relies on the combination of isotope and geo-data (e.g. assessed recharge areas (section 4.7), inferred 3D flow distance and ITTP). The statements and interpretation based on observed relationship between EC (and T) and flow distance were withdrawn and revised by solely mention that a similar pattern of increasing flow path length vs. decreasing discharge elevation was observed by increasing EC values vs. decreasing discharge elevation. With the available data and in the absence of a complete chemical characterisation of the water bodies the results from the (multitemporal) isotope analysis and their derivatives (recharge area, transit time, inferred flow path length) are even more important and require a comprising discussion as presented in section 5.

*In conclusion,*
*This is an interesting article that develops a methodology for estimating recharge as a function of water infiltration altitude using O18 isotopes and a numerical field model that allows us to constrain these recharge areas.*
*I think that the flow part still lacks data, especially on the different water masses. The only measurement of the electrical conductivity is not sufficient to constrain these flows and cannot in any case be used as a tracer of transit time in complex geological environments.*
*A suggestion for your next study*
*I would start by mapping the water outlets (temperature, pH (the pH would allow to isolate the flows on the silicates) and electrical conductivity) that I would place on a geological support (look at where these marble banks are in relation to the outlets).*
*Then I would select springs which are differentiated by these parameters and make high frequency measurements of electrical conductivity, water height (or flow) on a transect.*
*I would also equip the boreholes with this type of probe (does the bottom of the boreholes reach the fracture surface? If so, it's good because these surfaces are major drains of flows.*

*Nevertheless, I find your approach to determine the recharge zones really very good, your hydro study a little less so*

We appreciate the detailed feedback and revised the manuscript based on the data (and its derivatives) accordingly. Please contact us at any time in case we can provide the revised manuscript.

---

## Author Comment (AC2)

**Response to the Reviewers' comments**

(original comment in italics and response in green)

Review #2 (Anonymous)

https://doi.org/10.5194/nhess-2021-388-RC2, 2022

*The article describes interesting and possibly applicable method to evaluate hydrogeological conditions governing deep seated gravitational slope deformations. I found no major drawback of the presented article, while I think the method can be quite useful even for practical landslide mitigation purposes.*

We thank the reviewer for the positive feedback.